# Exploring the roles of roughness, friction and adhesion in discontinuous shear thickening by means of thermo-responsive particles

Chiao-Peng Hsu [1,2], Joydeb Mandal[2], Shivaprakash N. Ramakrishna[2], Nicholas D. Spencer [2] & Lucio Isa [1 ✉]

Dense suspensions of colloidal or granular particles can display pronounced non-Newtonian behaviour, such as discontinuous shear thickening and shear jamming. The essential contribution of particle surface roughness and adhesive forces confirms that stress-activated frictional contacts can play a key role in these phenomena. Here, by employing a system of microparticles coated by responsive polymers, we report experimental evidence that the relative contributions of friction, adhesion, and surface roughness can be tuned in situ as a function of temperature. Modifying temperature during shear therefore allows contact conditions to be regulated, and discontinuous shear thickening to be switched on and off on demand. The macroscopic rheological response follows the dictates of independent single-particle characterization of adhesive and tribological properties, obtained by colloidal-probe atomic force microscopy. Our findings identify additional routes for the design of smart non-Newtonian fluids and open a way to more directly connect experiments to computational models of sheared suspensions.

[1] Laboratory for Soft Materials and Interfaces, Department of Materials, ETH Zurich, Zurich, Switzerland. [2] Laboratory for Surface Science and Technology, Department of Materials, ETH Zurich, Zurich, Switzerland. ✉email: lucio.isa@mat.ethz.ch

**S**hear thickening (ST) is a generic phenomenon that occurs when the shear stress $\sigma$ increases faster than linearly with the shear rate $\dot{\gamma}$, so that the viscosity $\eta \equiv \sigma/\dot{\gamma}$ effectively increases with shear rate. It can be observed in a broad range of materials, but nowhere is it more prominent than in dense suspensions of solid particles[1,2]. This phenomenon may take the severe form known as discontinuous shear thickening (DST)[1,3–7], where the suspension's viscosity increases by orders of magnitude at a critical shear rate, or in the most extreme cases, the suspension may even solidify under shear—an occurrence known as shear jamming (SJ)[8–10]. Both instances can lead to failures in high-shear processes but can also be exploited for applications, e.g., in granulation or impact absorption. Recent studies have shown that interparticle contacts play a crucial role in DST, triggered by a change in the interactions between particle surfaces from hydrodynamic lubrication to boundary lubrication at high shear[5–7,11–15]. While fluid films allow suspended particles to easily slide past each other at low shear, beyond a critical shear stress the hydrodynamic lubrication films between particles break down. This occurrence results in particles that are effectively in asperity–asperity contact and thus can engage in frictional interactions via boundary lubrication. This transition indicates that the surface morphology and surface chemistry of the particles can have a striking influence on the macroscopic flow behavior. Based on this knowledge, an engineering-tribology approach can be utilized to design the thickening of suspensions. This approach offers strategies for the creation of energy-absorbing materials[16,17], medical devices[18], and flexible body armors[19–21]. The ability to predict and determine the onset of ST and its severity is also critical for the three-dimensional printing of composite inks[22] containing high solids loadings, such as in the case of ceramic or conductive inks carrying metallic microparticles, e.g., in the ink-jet printing of solder balls for the microelectronics industry[23].

From an experimental standpoint, interparticle friction coefficients can be effectively controlled by engineering surface chemistry[6] or by modifying surface roughness[24–26]. Moreover, short-range adhesive forces, e.g., as experimentally introduced by hydrogen bonding[27,28], can also be used to modify the properties of interparticle contacts and strongly affect the rheology. However, regulating the interparticle tribology while the suspensions are being sheared as a means to offer external control on the flow properties has not yet been addressed. External control of shear-thickening response has been realized, for instance, by mechanical triggers[29], i.e. vibrations, but an adaptation of the inter-particle interactions via easily tunable variables, such as temperature control in a printer's nozzle, presents practical advantages. Finally, exploring the different contributions that link the tribology of contacts to the rheology of the suspension remains an elusive task with important consequences for materials design and fundamental understanding alike.

In this work, we examine model silica colloids of varying surface roughness coated with thermo-responsive polymer brushes of poly(N-isopropylacrylamide) (PNIPAM). Comparing the nanotribology and rheology of these model colloids allows us to test the relative contributions of friction, adhesion, and surface roughness and to tune them during shear.

## Results

We synthesize raspberry-like silica particles with controllable roughness[26] and employ surface-initiated atom transfer radical polymerization (SI-ATRP)[30,31] to graft PNIPAM brushes from the particles' surfaces (see "Methods" and Supplementary Fig. 1). A distinctive feature of PNIPAM brushes is that they undergo a swelling–deswelling transition in water across a lower critical solution temperature (LCST) of 30–33 °C[32]. Figure 1 displays the different PNIPAM-grafted particles employed in this study. Smooth silica PNIPAM-grafted particles are named SM_PNIPAM and raspberry-like silica PNIPAM-grafted particles are named RB_$h/d$_PNIPAM, where $h/d$ is the value of the dimensionless roughness parameter of the silica surface measured before polymerization ($h$ is the average asperity height and $d$ is the average inter-asperity spacing obtained by atomic force microscopic (AFM) imaging of the particles' surfaces)[26]. $h/d$ directly reflects the linear density of asperities with height $h$ and can be used to compare roughness between the PNIPAM-grafted particles investigated in this work since they have the same core particle diameter and all asperities have a spherical cap morphology. The dehydration of PNIPAM brushes across the LCST results in different effective particle sizes as a function of temperature. In particular, we designed the swollen thickness of the PNIPAM brushes, $h_{PNIPAM}$, to ensure that the underlying roughness can be masked at 20 °C and revealed at 40 °C, as schematically shown in Fig. 1a. Moreover, the thickness of the collapsed polymer brush is also sufficient to mask the topography of the raspberry-like particles with the lowest roughness, i.e., RB_0.22_PNIPAM. Using SI-ATRP offers exquisite control over the brush thickness by selecting the precise solvent ratio, catalyst ratio, and polymerization time, hence enabling unique tailoring of the physico-chemical properties of model, brush-stabilized particles. The properties of the PNIPAM-grafted particles are summarized in Table 1.

The solubility transition of PNIPAM brushes not only affects their thickness but also has a strong impact on their surface interactions. In order to characterize this effect at the level of

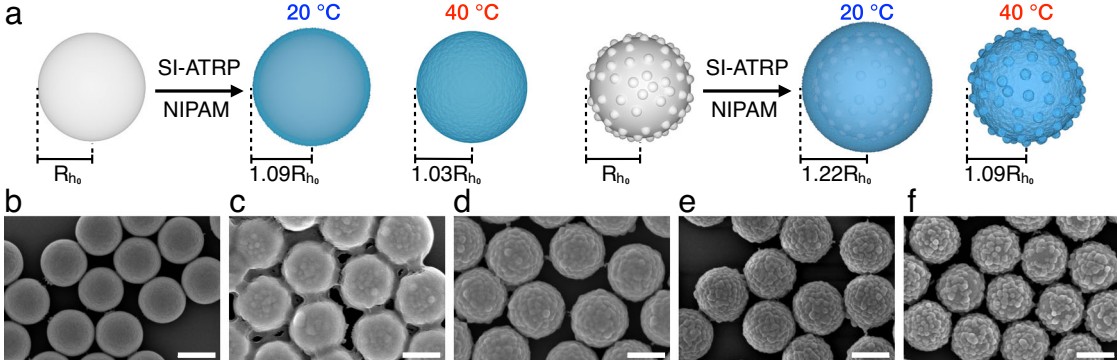

**Fig. 1 PNIPAM-grafted smooth and rough silica particles. a** Schematics of the PNIPAM-grafted smooth (SM) and rough (RB) particles at 20 °C and 40 °C. $R_{h_0}$ is the radius of smooth or rough particles before polymerization. **b-f** Scanning electron microscopic images of four PNIPAM-grafted particle systems: SM_PNIPAM (**b**), RB_0.22_PNIPAM (**c**), RB_0.36_PNIPAM (**d**), RB_0.46_PNIPAM (**e**), RB_0.52_PNIPAM (**f**). Scale bars indicate 500 nm.

**Table 1 Details of the PNIPAM-grafted particles.**

| | SM_PNIPAM | RB_0.22_PNIPAM | RB_0.36_PNIPAM | RB_0.46_PNIPAM | RB_0.52_PNIPAM |
|---|---|---|---|---|---|
| $h/d$ | $\approx 0$ | $0.22 \pm 0.03$ | $0.36 \pm 0.04$ | $0.46 \pm 0.05$ | $0.52 \pm 0.04$ |
| $R_{h_0}(20\,^\circ\text{C})$ (nm) | $339 \pm 4$ | $376 \pm 6$ | $385 \pm 5$ | $383 \pm 6$ | $379 \pm 5$ |
| $R_h$ (20 °C) (nm) | $370 \pm 4$ | $466 \pm 6$ | $469 \pm 6$ | $468 \pm 7$ | $459 \pm 6$ |
| $R_{h_0}(40\,^\circ\text{C})$ (nm) | $339 \pm 3$ | $381 \pm 7$ | $388 \pm 5$ | $388 \pm 5$ | $385 \pm 4$ |
| $R_h$ (40 °C) (nm) | $351 \pm 3$ | $419 \pm 6$ | $423 \pm 5$ | $421 \pm 5$ | $416 \pm 5$ |
| $R_h(20\,^\circ\text{C})/R_{h_0}(20\,^\circ\text{C})$ | $1.09 \pm 0.02$ | $1.23 \pm 0.03$ | $1.22 \pm 0.02$ | $1.23 \pm 0.03$ | $1.22 \pm 0.02$ |
| $h_{\text{PNIPAM}}$ (20 °C) (nm) | $31 \pm 6$ | $90 \pm 8$ | $84 \pm 8$ | $85 \pm 9$ | $80 \pm 8$ |
| $R_h(40\,^\circ\text{C})/R_{h_0}(40\,^\circ\text{C})$ | $1.03 \pm 0.01$ | $1.10 \pm 0.03$ | $1.08 \pm 0.02$ | $1.09 \pm 0.02$ | $1.08 \pm 0.02$ |
| $h_{\text{PNIPAM}}$ (40 °C) (nm) | $12 \pm 4$ | $38 \pm 8$ | $35 \pm 7$ | $33 \pm 7$ | $31 \pm 6$ |
| $m_{\text{PNIPAM}}/m_{\text{SiO}_2}$ | 0.11 | 0.14 | 0.13 | 0.13 | 0.12 |

$h/d$ is the dimensionless roughness parameter. $R_h$ $(T)$ and $R_{h_0}(T)$ are the particle hydrodynamic radii after and before polymerization, respectively, at the given temperature. $h_{\text{PNIPAM}}$ $(T)$ is the PNIPAM thickness, for which $h_{\text{PNIPAM}}$ $(T) = R_h$ $(T) - R_{h_0}$ $(T)$. $m_{\text{PNIPAM}}/m_{\text{SiO}_2}$ is the mass fraction of PNIPAM brushes and silica particles.

nanoscale contacts, we measure the tribological properties of PNIPAM-grafted particles by colloidal-probe AFM. In particular, we provide controlled contact conditions by attaching individual PNIPAM-coated particles of a given roughness onto an AFM cantilever and sliding them against a flat, PNIPAM-coated countersurface of the same underlying roughness in water at two temperatures—above (40 °C) and below the LCST (20 °C), respectively—measuring friction as a function of the applied load. The substrates are produced by a process analogous to the synthesis of the polymer-coated colloids, to provide representative, realistic countersurfaces mimicking interparticle contacts in the sheared suspensions (see "Methods" and Supplementary Fig. 2).

Crossing the LCST changes both the adhesion and the friction forces between PNIPAM-coated surfaces[33]. To decouple these contributions from particle topography, we first examine the case of smooth particles (Fig. 2). We start by characterizing the adhesion force between PNIPAM-grafted particles at different temperatures by measuring the pull-off force in a force-versus-distance curve (see "Methods"). Representative force–distance curves at 20 and 40 °C are shown in Fig. 2a. The retraction curve below LCST shows no adhesion force. Conversely, above the LCST, a small jump to contact is observed during approach and the retraction curve displays marked adhesion with a pull-off force of $4.2 \pm 0.9$ nN as shown in Fig. 2b. This adhesion is due to the hydrophobic interaction between the collapsed PNIPAM brushes[34,35]. We then measure the friction by means of lateral force microscopy (see "Methods"). The friction forces above the LCST are ~20 times higher than below the LCST, as shown in Fig. 2c, and a non-zero friction force at zero applied load is found, in agreement with the presence of adhesion at the contact. At 20 °C, the polymer chains are stretched away from the surface into a brush-like conformation. The combination of osmotic pressure within the grafted assemblies and steric resistance to interpenetration between opposing brushes leads to low friction when sliding against each other. At 40 °C, the chains are dehydrated into a collapsed conformation, which presents a much lower osmotic and steric repulsion between the opposing surfaces, resulting in non-lubricious sliding. The hydrophobic interactions between the collapsed chains of the opposing surfaces also cause more dissipation while sliding, due to adhesion hysteresis[36]. Moreover, at 20 °C, the SM_PNIPAM particles and smooth silica particles without the polymer (SM) display almost identical low-friction behavior at applied loads below 30 nN. At higher applied loads, the SM_PNIPAM particles start to show higher friction forces than SM particles. Further details of the surface interactions were obtained by measurement of the friction force versus sliding distance in the so-called "friction loops" (Fig. 2cI–IV). The narrow

loop for SM_PNIPAM at 30 nN (Fig. 2cI, II) is indicative of smooth sliding and low friction. Upon increasing the applied load to 60 nN (Fig. 2cIII, IV), the friction loop opens up, indicating increased dissipation, which has been ascribed to the entanglement of PNIPAM brushes at higher loads[37]. The friction loops at 40 °C always show higher dissipation and irregular sliding, as a consequence of adhesion.

In the presence of adhesion force, the relation between friction force $F_{\text{friction}}$ and applied load $L$ in Fig. 2c allows a friction coefficient to be extracted using a modified version of Amontons' Law[38]:

$$F_{\text{friction}} = \mu \cdot (L_0 + L) = F_0 + \mu \cdot L, \tag{1}$$

where a constant internal load $L_0$ is added to the applied load $L$, in order to account for the intermolecular adhesive forces. $F_0$ represents the friction force at zero applied load for adhesive surfaces. The friction coefficient $\mu$ is then defined as the slope $dF_{\text{friction}}/dL = \mu$, which we plot in Fig. 3a (top).

Several important observations can be made when comparing the tribological and rheological behavior of our particles as a function of temperature.

Our previous works have shown that the interparticle friction coefficient is directly correlated to the high-shear maximum packing fraction obtained in centrifugation experiments $\phi_m$ (see "Methods")[6,26,39]. We also observe here that a system having higher $\mu$ has a lower $\phi_m$ (Fig. 3a—bottom). However, we notice that, although the $\mu$ values of SM_PNIPAM and SM are fairly close at 20 °C, there is ~10% difference in their $\phi_m$ values. This discrepancy is due to the fact that SM_PNIPAM has a soft and compressible shell, resulting in a denser packing during sedimentation and highlights that the detailed relationship between $\mu$ and $\phi_m$ may be system specific.

The distinct adhesion and friction behaviors of SM_PNIPAM at 20 °C and 40 °C also lead to distinct shear-rheology responses as a function of temperature. In Fig. 3b, we examine the flow curves of three aqueous dense suspensions of SM_PNIPAM. Because of the swelling and dehydration of the PNIPAM brushes, each suspension has a different effective $\phi$ at different temperature, e.g., $\phi = 64\%$ at 20 °C and $\phi = 54\%$ at 40 °C for the same suspension. However, qualitative differences beyond a rescaling of the volume fraction emerge. In particular, clear variations in the flow curves are noted if we rescale the volume fraction with respect to $\phi_m$ for the different systems, introducing $\Delta\phi^* = (\phi_m - \phi)/\phi_m$. This rescaling specifically shows the difference between the volume fraction $\phi$ at which the rheology is measured and the corresponding value of $\phi_m$. It has recently been shown that this rescaled quantity can be correlated with the strength of ST in colloidal suspensions[40]. SM_PNIPAM colloids at 20 °C start

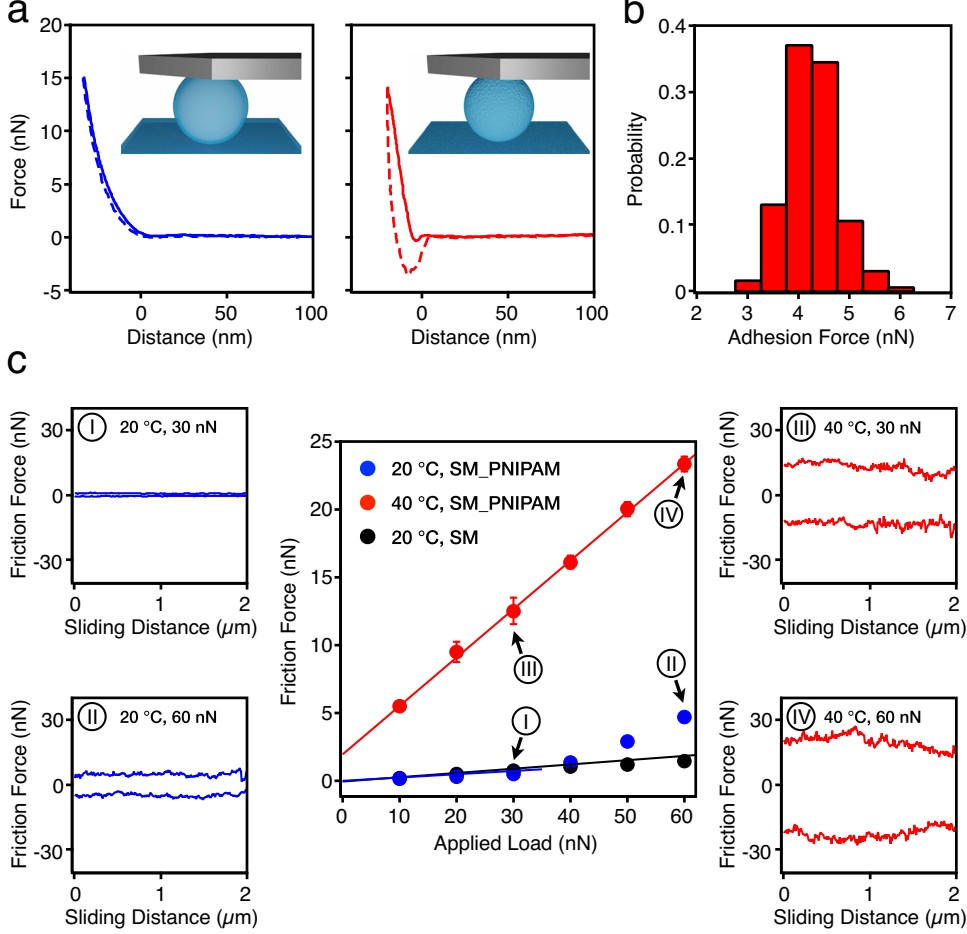

**Fig. 2 Nanotribology experiments on PNIPAM-grafted smooth particles. a** Approach (solid line) and retraction (dashed line) of the force–distance curves of SM_PNIPAM system at 20 °C (left) and 40 °C (right). The zero-point is determined as the point at which the approach curve shows non-zero vertical deflection due to interaction with the substrate. The top two insets show the schematics of a SM_PNIPAM probe on a countersurface at 20 °C (left) and 40 °C (right). **b** The adhesion forces measured at 40 °C. The bin width is 0.5 nN. **c** Friction force versus applied load measurements of SM_PNIPAM at 20 °C (blue) and 40 °C (red) and SM (black). Error bars represent the standard deviations across the scan area (51 data points). Friction loops of the SM_PNIPAM system at 30 nN applied load and 20 °C (I); at 60 nN and 20 °C (II); at 30 nN and 40 °C (III); at 60 nN and 40 °C (IV).

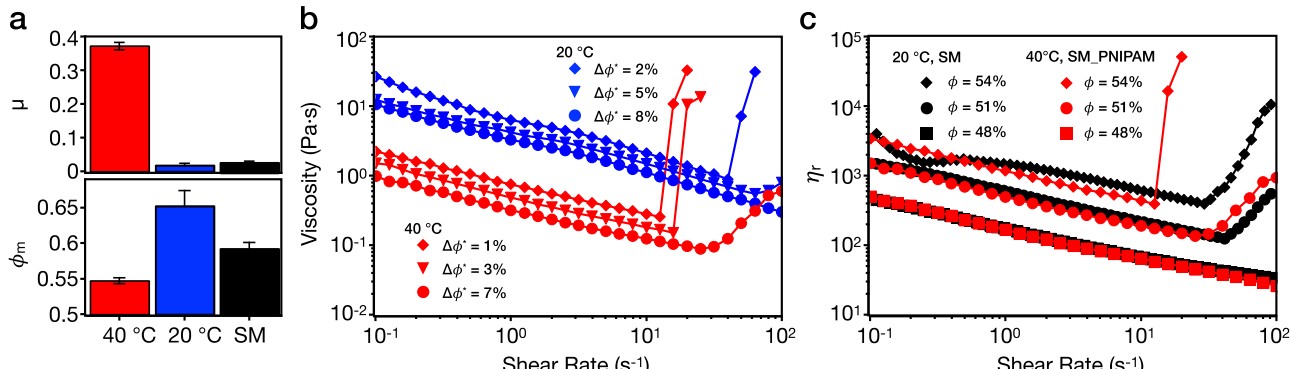

**Fig. 3 Rheology experiments on PNIPAM-grafted smooth particles. a** Friction coefficients $\mu$ (top) and frictional packing fraction $\phi_m$ (bottom) of the SM_PNIPAM system at 40 °C (red) and 20 °C (blue) and SM system (black). Error bars in $\mu$ represent the uncertainties in the calculated slope based on Eq. (1). $\mu$ of SM_PNIPAM at 20 °C was calculated from 10 to 30 nN. Error bars in $\phi_m$ represent the standard deviations from four repeat measurements. **b** Rheological flow curves of SM_PNIPAM in rescaled packing fractions $\Delta\phi^*$ at 20 °C (blue) and 40 °C (red). **c** Normalized flow curves of SM_PNIPAM at 40 °C (red) and SM at 20 °C (black) for $\phi = 48$, 51, and 54%. The relative viscosity $\eta_r$ is calculated based on $\eta_r = \eta_{measured}/\eta_{water}$ (T). The results for the SM system are adapted from previous work[26].

displaying continuous shear thickening (CST) for $\Delta\phi^* = 5\%$ and exhibit DST at $\Delta\phi^* = 2\%$—very close to their measured $\phi_m$ of 65.2% at 20 °C. At 40 °C, adhesion and higher friction switch on, which shift the onset of ST to lower shear rates and enhance its magnitude. The same suspension exhibits CST at 40 °C ($\Delta\phi^* = 7\%$), while showing shear thinning at 20 °C ($\Delta\phi^* = 8\%$). At higher solid loading, the same suspension transitions between CST at 20 °C ($\Delta\phi^* = 5\%$) to DST at 40 °C ($\Delta\phi^* = 3\%$). For the densest suspension where dilatant DST is observed at both temperatures (positive $N_1$, see Supplementary Fig. 3a), the onset rate and stress (Supplementary Fig. 3b) occur at significantly lower values at 40 °C. In our system, the brush-like conformation of the swollen PNIPAM leads to steric repulsion at 20 °C, which keeps the silica surfaces separated until a critical load is exceeded and brush interpenetration causes an increase in dissipation. Residual charges on the silica surface offer an electrostatic repulsive force at 40 °C when the PNIPAM brushes are deswollen.

To isolate the effect of volume fraction from that of contact properties, it is worth comparing the flow curves of low-frictional and non-adhesive SM colloids without PNIPAM at 20 °C with the high-frictional and adhesive SM_PNIPAM colloids at 40 °C at the same $\phi$ values (Fig. 3c). Strikingly, for the shear-thinning and CST regimes with $\phi$ up to 51%, both suspensions behave very similarly, even if the contact properties are radically different, confirming that the hydrodynamic volume fraction is the dominating parameter in describing the rheology. This observation is consistent with the results from classical Stokesian dynamics simulations[41,42] which show that CST in dense suspensions can be generated by hydrodynamic forces without allowing surface contact. Moreover, in our case CST is coupled with a negative $N_1$ (Supplementary Fig. 3c), which confirms that hydrodynamic effects are dominant. Conversely, a marked difference is observed concerning the DST behavior. At $\phi = 54\%$, SM_PNIPAM and SM display similar low-shear rheology, but at high shear rates the former exhibits DST and the latter shows CST. This finding reinforces the view that contact properties strongly affect DST over and above hydrodynamic effects.

Moving from SM to RB particles, additional features emerge. The RB_PNIPAM systems also display switchable adhesion across the LCST (Fig. 4a and Supplementary Fig. 4), with an adhesion force of 5.3 ± 1.3 nN between RB_0.52_PNIPAM and its countersurface at 40 °C (Fig. 4b). This value is not very different from that measured for the SM_PNIPAM, confirming the analogy in the chemical origin of the adhesive bonds. As opposed to the smooth particles' case, the precise control of brush thickness allows us to switch on and off the underlying surface roughness of the RB_PNIPAM particles at different temperatures. This switching has a striking influence on the friction behavior of the RB_PNIPAM systems, as shown in Fig. 4c. At 20 °C, all four RB_PNIPAM systems behave like SM_PNIPAM and SM, exhibiting identical and low friction below 30 nN. This fact indicates that the swollen PNIPAM shell effectively screens the roughness at low applied loads, as borne out by the flat, narrow friction loop for RB_0.52_PNIPAM at 30 nN (Fig. 4cI). Upon increasing the applied load, i.e., to 60 nN, fluctuations in the friction loop appear due to the interlocking of asperities, which are now revealed as the polymer brush is highly compressed, even below the LCST (Fig. 4cII). Conversely, at 40 °C, the RB_PNIPAM systems show high, roughness-dependent friction, with higher friction forces measured for particles with higher underlying roughness (Fig. 4c).

An exception is the case of the RB_0.22_PNIPAM system, where the polymer brush is always able to mask the underlying surface roughness, both at high and low temperature and at low and high applied load. This can be clearly seen from the friction loops at 20 and 40 °C, for a 30- and a 60-nN applied load (Fig. 4I–IV). In all these conditions, smooth friction loops are

seen for the RB_0.22_PNIPAM system in contrast to the RB_0.52_PNIPAM system, which only shows a smooth loop at low temperatures and low applied loads, as described above. Moreover, at 40 °C, the RB_0.22_PNIPAM system shows the lowest friction forces compared to other RB_PNIPAM systems. Essentially, three factors contribute to the increase of friction above the LCST: the collapse of the brushes, which eliminates the osmotic and steric repulsion, the adhesion between the PNIPAM brushes, and the unveiling of roughness. Above the LCST, the surface of the RB_PNIPAM particles turns sticky and rough because of the collapse of PNIPAM brushes, allowing asperities to interlock within sticky contacts. At 40 °C, we thus observe the wide friction loops of RB_0.52_PNIPAM system with stick–slip events at all loads, as shown in Fig. 4cIII, IV. One can further compare the friction loops at 40 °C and 60 nN applied load for all the PNIPAM-grafted systems (Supplementary Fig. 5). It is clear that the RB_0.36_PNIPAM, RB_0.46_PNIPAM, and RB_0.52_PNIPAM particles show large oscillations in the lateral force caused by underlying surface asperities, while the RB_0.22_PNIPAM particles behave very similarly to the SM_PNIPAM particles.

As a consequence of the differences in the frictional behavior at the different temperatures for the four RB_PNIPAM particles studied, a clear relation between $\phi_m$ and $\mu$ emerges, as shown in Fig. 5a. Confirming previous observations, we see that the higher the effective interparticle friction coefficient, the lower the value of $\phi_m$[6,26]. Because the effect of topography emerges only above the LCST, the differences between the three systems emerge only at higher temperatures. Conversely, our particles show almost identical friction coefficient and $\phi_m$ values below the LCST, again supporting the conclusion that swollen PNIPAM shells effectively screen roughness. As already introduced for the smooth particles, the temperature-dependent tribology of the rough colloids strongly affects their rheology at high packing fractions, as shown for aqueous dense suspensions of RB_0.52_PNIPAM in Fig. 5b. At 20 °C, the suspensions do not display any appreciable CST for $\Delta\phi^*$ up to 7% but directly show DST at 4%. This behavior is analogous to the previously reported ST response of uncoated, rough silica spheres[26] and stems from the interlocking of asperities when a high shear stress breaks through the PNIPAM lubrication layer, as clearly shown by the presence of fluctuations in the friction loop at high load (Fig. 4cII). The same suspensions display a distinctively different response at 40 °C. All three suspensions show dilatant DST (positive $N_1$, see Supplementary Fig. 6a), and the onset rate for DST shifts to lower shear rates and shear stresses (Supplementary Fig. 6b) with increasing $\phi$. DST is even observed for values of $\Delta\phi^*$, for which the suspension below the LCST only shows shear thinning, i.e., $\Delta\phi^* = 7\%$. The strongly enhanced DST above the LCST is generated by the sticky, rough particle surface, where in addition to asperity interlocking, adhesion between the collapsed PNIPAM brushes causes a significant increase in the effective friction.

Finally, we compare the shear rheology of the four RB_PNIPAM systems at the same $\phi$ as shown in Fig. 5c. The flow curves obtained below the LCST with $\phi = 63\%$ are all identical and show shear thinning, implying that with no ST the rheology is defined by the packing fraction alone. Above the LCST, surface differences emerge only during ST, whose onset becomes roughness dependent. Below the onset of DST, all suspensions, which have also the same $\phi$ at 40 °C, show the same viscosity, but the different roughness causes the breakdown of the fluid lubrication layers at different stresses, leading to DST taking place at lower shear rates for higher roughness. In other words, at the same $\phi$, the existence of differences in $\mu$ and $\phi_m$ are hidden until the DST onset and the smaller the value of $\Delta\phi^*$, the lower are the observed critical shear rate and shear stress (Supplementary Fig. 6). It is

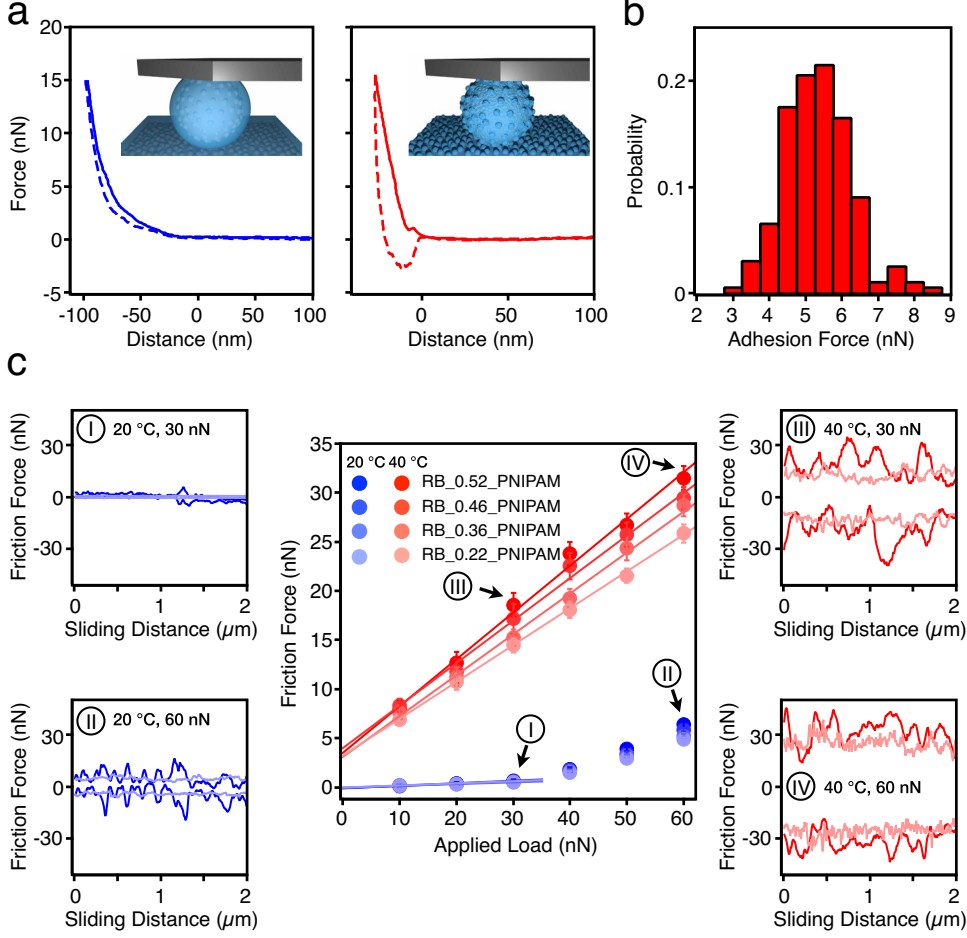

**Fig. 4 Nanotribology experiments on PNIPAM-grafted rough particles. a** Approach (solid line) and retraction (dashed line) of the force–distance curves of RB_0.52_PNIPAM system at 20 °C (left) and 40 °C (right). The zero-point is determined as the point at which the approach curve shows non-zero vertical deflection due to interaction with the substrate. The top two insets show the schematics of a RB_0.52_PNIPAM probe on a countersurface at 20 °C (left) and 40 °C (right). **b** The adhesion forces measured at 40 °C. The bin width is 0.5 nN. **c** Friction force versus applied load measurements of RB_0.22_PNIPAM, RB_0.36_PNIPAM, RB_0.46_PNIPAM, and RB_0.52_PNIPAM at 20 °C (blue) and 40 °C (red). Error bars represent the standard deviations across the scan area (51 data points). Friction loops of the RB_0.22_PNIPAM and RB_0.52_PNIPAM system at 30 nN applied load and 20 °C (I); at 60 nN and 20 °C (II); at 30 nN and 40 °C (III); at 60 nN and 40 °C (IV).

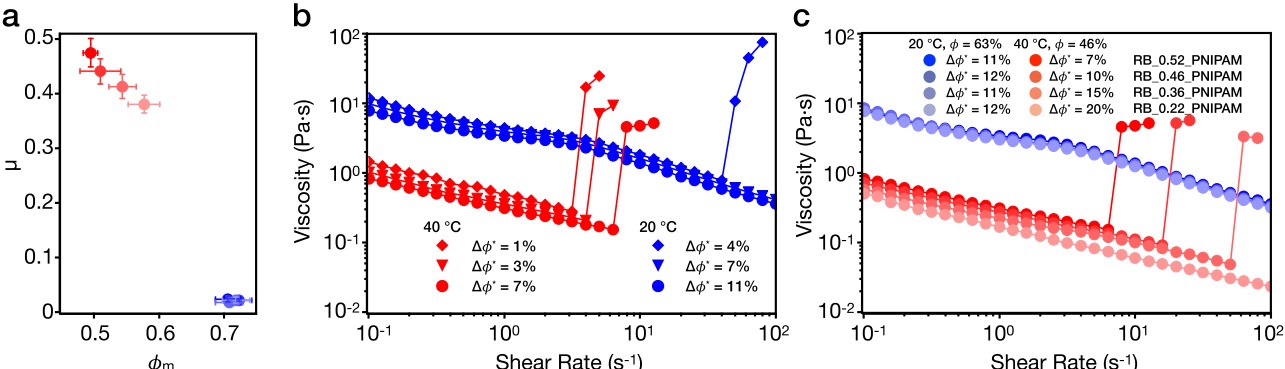

**Fig. 5 Rheology experiments on PNIPAM-grafted rough particles. a** $\mu$ as a function of $\phi_m$ of the four PNIPAM-grafted rough systems (same legends as Fig. 4c) at 20 °C (blue) and 40 °C (red). $\mu$ at 20 °C was calculated from 10 to 30 nN applied load. Error bars in $\mu$ and $\phi_m$ represent the uncertainties in the calculated slope based on Eq. (1) and the standard deviations from four repeat measurements, respectively. **b** Rheological flow curves of RB_0.52_PNIPAM in various values of $\phi$ at 20 °C (blue) and 40 °C (red). **c** Rheological flow curves of the four PNIPAM-grafted rough systems with $\phi = 63\%$ at 20 °C (blue) and $\phi = 46\%$ at 40 °C (red).

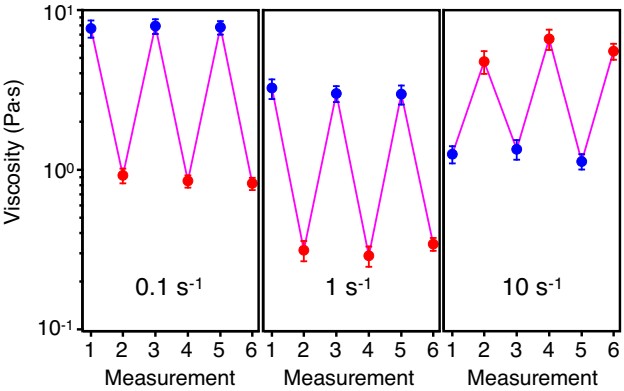

**Fig. 6 Thermo-switchable shear-thickening system.** Viscosity measurements at shear s of $0.1\,s^{-1}$ (left), $1\,s^{-1}$ (middle), and $10\,s^{-1}$ (right) of RB_0.52_PNIPAM at 20 °C; $\phi = 63\%$ (blue) and 40 °C; $\phi = 46\%$ (red). Error bars represent the standard deviations from ten repeat measurements.

worth pointing out that RB_0.22_PNIPAM shows the lowest $\phi_m$ and the roughness is still masked above the LCST. As a consequence, RB_0.22_PNIPAM shows pure shear thinning, while the other shows DST at the same volume fraction.

From these results, a clear picture emerges in which the PNIPAM brushes effectively tune the friction, adhesion, and surface roughness of RB_PNIPAM particles in suspension as a function of temperature. Essentially, the swelling of the brushes can mask surface roughness, reduce friction, and switch off adhesion, so that the DST of the dense suspensions can be significantly retarded. The collapse of the brushes instead introduces adhesive forces, increases friction, and unveils surface roughness so that DST is enhanced. Naturally, changing the temperature across the LCST also causes a shift of the suspension's volume fraction, due to the temperature-dependent particle size. Measurements of the hydrodynamic radii of the particles at 20 and 40 °C over multiple cycles indicate full reversibility of the particle swelling–deswelling and show that particles remain colloidally stable even at high temperatures (Supplementary Fig. 7). These results have inspired us to utilize the PNIPAM-grafted colloids as a thermo-switchable shear-thickening system, as shown in Fig. 6. Because of the difference in $\phi$, at shear rates of 0.1 and $1\,s^{-1}$, below the DST onset at 40 °C, the same dense suspension of RB_0.52_PNIPAM exhibits a lower viscosity at 40 °C than at 20 °C. Nonetheless, the situation is inverted at $10\,s^{-1}$, where the suspension shows DST at 40 °C and shear thinning at 20 °C. Given the rapid response of PNIPAM to temperature changes[43], DST can effectively be switched on and off on demand. By cycling the temperature up and down during shear, we show that the viscosity switch is reversible because of the reversibility of the swelling–deswelling transition of PNIPAM brushes. Cycling the temperature above and below the PNIPAM's LCST enables the complete recovery of the suspension. Similarly, repeating flow curves at 40 °C shows an identical response, indicating that suspensions entering the DST regime can be fully rejuvenated by shear (Supplementary Fig. 8).

## Discussion

Summarizing, we have presented a unique approach to use stimuli-responsive polymer brushes, synthesized to a precise length by controlled radical polymerization, for the in situ tuning of interparticle friction and adhesion and for the modulation of surface roughness. Our previous results demonstrated that, in the absence of adhesion, the interlocking of surface asperities promoted dilatant DST over a broader range of shear rates and for

lower values of $\phi$ with increasing (roughness-mediated) effective friction[26]. These experiments additionally show that introducing weak adhesive forces also strengthens ST and shifts it to lower shear rates for smooth contacts. The combination of both effects exacerbates DST for rough adhesive particles.

This phenomenology presents strong links to recent numerical work, which proposes a constraint-based approach to ST[44–46]. In addition to resisting sliding within shear-induced contact regions as a consequence of friction and topography, adhesion introduces further constraints to the rolling between particles[47], which effectively lowers the high SJ volume fraction, thus promoting ST at a given $\phi$.

Although friction coefficients and adhesive forces can be independently tuned in a numerical simulation[5,15,46] to regulate $F_{friction}$, their complete decoupling in experiments is essentially not practically possible. It has in fact been well documented that solids with high adhesion normally have a higher friction force both in macroscale as well as in nanoscale contacts[48–52]. At the nanoscale, an increase in the adhesion changes the real area of contact, which can potentially alter the entire contact mechanical behavior[51,53,54]. More specifically, as the friction force is assumed to be directly proportional to the real area of contact, experimentally, it is hard to think about a system in which the adhesive force changes but the real area of contact does not. The connection between these two factors therefore implies that a complete, orthogonal disentanglement between friction and adhesion cannot be reached in a real system. However, our experimental results indicate that the relative contributions of roughness, friction, and adhesion can be modified in one single system by exploiting responsive polymer brushes. Our findings therefore encourage future simulation studies to include realistic constraints and relationships between sliding and rolling friction coefficients and adhesive forces. Moreover, we present a route where the variations of the contact properties are reversible and externally controlled during shear and do not require a modification of the composition of the materials, i.e., via the addition of surfactants[45,55] and small molecules (e.g., urea)[27].

Concluding, our results clearly confirm that the microscopic tribology of interparticle contacts is strongly associated with the macroscopic rheology of DST suspensions. As a consequence, additional routes for designing shear-thickening materials via tuning the particle surface chemistry and morphology open up, in particular developing responsive systems in which friction, adhesion, and surface roughness can be engineered on demand.

## Methods

**Materials.** Silica particles 8 nm (Nyacol), 22 nm (Ludox® TM; DuPont), and 600 nm (Nanocym), aminopropyl-functionalized silica particles 650 nm (Nanocym), polydiallyldimethylammonium chloride solution (poly-DADMAC, 400–500 kDa, 20 wt% in water; Sigma-Aldrich), tetraethyl orthosilicate (TEOS; 98%; Sigma-Aldrich), ammonia (25% in water; Sigma-Aldrich), (3-aminopropyl)triethoxysilane (APTES) (97%; Sigma-Aldrich), α-bromoisobutyryl bromide (BIBB) (99%; Sigma-Aldrich), triethylamine (99.5%, Sigma-Aldrich), copper(II) bromide (CuBr₂) (99%; Sigma-Aldrich), tetraethylammonium bromide (TEAB) (98%; Sigma-Aldrich), tris [2-(dimethylamino)ethyl]amine (Me₆TREN) (97%; Sigma-Aldrich), polyethyleneimine (PEI) (branched, high molecular weight; Sigma-Aldrich), ethanol (99.8%, extra dry; ACROS Organics), dichloromethane (DCM) (99.8%, extra dry; ACROS Organics), silicon wafers (10 mm × 10 mm; Si-Mat Silicon Wafers), capillary tubes (5015; Vitrotubes), and ultraviolet (UV)-curing glue (Norland Optical Adhesive 63; Norland Products) were used as received. N-isopropylacrylamide (NIPAM) (97%; Sigma-Aldrich) was purified by crystallizing it from a toluene/hexane (3:2, v/v) mixture and was dried under vacuum prior to use. Copper(I) bromide (CuBr) (99.99%; Sigma-Aldrich) was purified by stirring overnight in glacial acetic acid, filtering, and washing several times with acetone and diethyl ether and was dried under vacuum prior to use.

**Synthesis of raspberry-like silica particles.** The synthesis of rough particles was modified from a previous procedure[26]. In this work, 600-nm silica particles were used as the cores and 8- and 22-nm silica particles were used as the berries. In a typical procedure, a total of 1 g core particles and 625 μL of poly-DADMAC were

dispersed in 600 mL of Milli-Q water. This aqueous suspension was stirred at 750 rpm for 40 min in order to obtain positively charged core particles. The core particles were then washed three times (1055 rcf, 10 min), and the aqueous supernatant was exchanged with Milli-Q water to remove the excess of poly-DADMAC. Afterwards, the 1 g cleaned core particles were dispersed in 300 mL of Milli-Q water. Either 10 mL of 8-nm particles (1 wt%) or 20 mL of 22-nm particles (1 wt%) were added to the suspension under stirring at 500 rpm for 60 min. The obtained raspberry-like particles were washed three times by centrifugation (1055 rcf, 10 min), and the aqueous supernatant was exchanged with Milli-Q water. After the final wash, the cleaned raspberry-like particles were dispersed in 100 mL of Milli-Q water, giving a 1-wt% suspension.

TEOS molecules were used to grow an amorphous silica layer on the surface of the raspberry-like particles. A total of 74.4 mL ethanol, 12.2 mL ammonia, and 10 mL 1-wt% raspberry-like particle suspension were mixed. A 5 vol% of TEOS in ethanol solution was injected into the mixture by a syringe pump with a rate of 2 mL/h while sonicating. The TEOS solution was added in cycles. Each cycle consisted of adding 0.267 mL of TEOS solution following by an additional 25-min break. The reaction temperature was kept constant by applying water cooling. In case of using 8-nm particles, 9 cycles of smoothing process were applied. In case of using 22-nm particles, different cycles (6, 9, and 12) of the smoothing process were applied to tune the particles' surface roughness. The smoothed raspberry-like particles were washed three times (1055 rcf, 10 min), and the aqueous supernatant was exchanged with Milli-Q water. After the final wash, the cleaned particles were dispersed in a 1-wt% suspension. The smoothing experiments were repeated in order to smooth total 1 g raspberry-like particles for each batch.

**Immobilization of SI-ATRP initiators.** The surfaces of the rough particles were activated and functionalized with APTES in order to graft the initiators for polymerization. 500 mg of rough particles were mixed with 1 mL of APTES in 100 mL of dry ethanol for 12 h under stirring. The APTES-functionalized rough particles were cleaned three times with ethanol (1055 rcf, 10 min) and dried under vacuum. 500 mg of APTES-functionalized rough particles or aminopropyl-functionalized 650-nm smooth particles were dispersed in 50 mL of dry DCM after degassing with nitrogen for 30 min. The functionalized particles were then reacted with 2 mL of BIBB and 4 mL of triethylamine under nitrogen atmosphere for 4 h. The initiator-grafted particles were cleaned three times (1055 rcf, 10 min) with DCM and dried prior to use.

**Synthesis of PNIPAM brushes on silica particles.** SI-ATRP was used to graft the PNIPAM brushes on silica particles. The reaction scheme is illustrated in Supplementary Fig. 1. 300 mg of initiator-grafted particles were dispersed in 13 mL of ethanol/water (4:1, v/v) mixture and degassed with nitrogen for 1 h. A 25-mL ethanol/water (4:1, v/v) mixture containing 12 g (106 mmol) of NIPAM, 2.04 g (9.7 mmol) of TEAB, and 96 μL (0.35 mmol) of $Me_6TREN$ was degassed with nitrogen for 1 h and subsequently transferred using a degassed syringe to a flask containing 34.4 mg (0.24 mmol) of CuBr and 26.8 mg (0.12 mmol) of $CuBr_2$, kept under nitrogen. The solution was then stirred for 10 min until complete dissolution of the catalyst. The catalyst-containing solution was subsequently transferred using a degassed syringe to the particle solution to carry out the polymerization under nitrogen atmosphere for 30 min. The reaction was quenched by adding 40 mL of ethanol/water (4:1, v/v) mixture and by exposing the reaction mixture to air. The PNIPAM-grafted particles were cleaned three times with Milli-Q water (1055 rcf, 10 min) and kept in Milli-Q water for further use.

**Characterization of PNIPAM brushes on silica particles.** Dynamic light scattering (DLS) (Zetasizer Nano ZS; Malvern Panalytical) experiments were performed at 20 and 40 °C in Miilli-Q water to determine the particles' size. The thickness of PNIPAM brushes $h_{PNIPAM}$ were obtained by comparing the particles' hydrodynamic radii before ($R_{h_0}$) and after ($R_h$) the polymerization. Thermogravimetric analysis (TGA) (TGA/DSC 3+; METTLER TOLEDO) was used to determine the mass fraction of PNIPAM brushes $m_{PNIPAM}$ and silica particles $m_{SiO_2}$ of the PNIPAM-grafted particles: $m_{PNIPAM}/m_{SiO_2} = (m_{850°C} - m_{250°C})/m_{850°C}$ (Supplementary Fig. 9). The sample was heated up to 80 °C at 10 °C/min and kept at 80 °C for 1 h to remove the water. The sample was then heated up to 900 °C at 10 °C/min.

**Fabrication of rough substrates.** The particle-coated silicon wafers were fabricated according to previous work[26]. The silicon wafers were cleaned by ultrasonication for 10 min in ethanol and then 10 min in Milli-Q water, followed by 20 min UV/ozone cleaning (ProCleaner PLUS; UVFAB). The cleaned silicon wafers (~12–15 pieces) were then immersed in 1 mg/mL PEI solution for 30 min. After the adsorption of PEI, the silicon wafers were rinsed with Milli-Q water and dried with a nitrogen jet. PEI-coated silicon wafers were immersed in 0.002 wt% of 8-nm silica particle suspensions or 0.004 wt% of 22-nm silica particle suspensions for 20 min. Afterwards, the samples were removed from the suspension and rinsed with Milli-Q water.

For the smoothing process, 7.44 mL ethanol, 1.22 mL ammonia, and 1 mL Milli-Q water were mixed in a polytetrafluoroethylene container together with the rough substrates (~4–6 pieces) and different amount of 1 vol% TEOS in ethanol solution

was added. For RS_0.22_PNIPAM, 0.9 mL of 1 vol% TEOS solution was added in the smoothing process. For RS_0.36_PNIPAM, RS_0.46_PNIPAM, and RS_0.52_PNIPAM, different surface roughness were achieved by adding different amounts (0.6, 0.9, and 1.2 mL) of 1 vol% TEOS in ethanol solution in the smoothing process. After the reaction, the rough substrates were rinsed with Milli-Q water and dried with a nitrogen jet.

**Synthesis of PNIPAM brushes on planar substrates.** The same protocol used for particles was applied to graft the PNIPAM brushes on smooth and rough substrates. The substrates (~12–15 pieces) were functionalized with APTES in dry ethanol and subsequently rinsed with ethanol and blown dry with a nitrogen jet. The APTES-modified substrates were then reacted with BIBB and triethylamine in dry DCM after degassing under nitrogen. The reaction was continued for 4 h and finally the substrates were cleaned with DCM and blown dry with a nitrogen jet. Instead of 300 mg of particles, 6 pieces of smooth or rough substrates were used in the SI-ATRP process. The reaction times for the smooth and rough substrates was 20 and 25 min, respectively. The difference in reaction time was to achieve PNIPAM brushes with thicknesses close to the PNIPAM-grafted smooth and rough particles. The PNIPAM-grafted substrates were cleaned with toluene and blown dry with a nitrogen jet. Smooth PNIPAM-grafted substrates are named SM_PNIPAM and rough PNIPAM-grafted substrates are named RS_h/d_PNIPAM, where h/d is the value of the dimensionless roughness parameter of the silica surface measured before polymerization. The SEM images are shown in Supplementary Fig. 2.

**Characterization of PNIPAM brushes on planar substrates.** The thickness of the PNIPAM brushes was measured using variable-angle spectroscopic ellipsometry (M-2000F; J. A. Woollam Co.). All data were recorded between 275 and 827 nm of wavelength at 70° with respect to the surface. The obtained values of amplitude (Ψ) and phase (Δ) components were analyzed with the WVASE32 software using a three-layer model (Si/SiO₂/Cauchy; $A_n = 1.45$, $B_n = 0.01$, and $C_n = 0$). All measurements were performed in Milli-Q water at 20 and 40 °C. The thickness of the rough SiO₂ layer on the substrate was measured before the polymerization and the value was used in the three-layer model to determine the PNIPAM thickness. The measured PNIPAM thicknesses are listed in Supplementary Table I.

**Centrifugation experiments.** The centrifugation experiments were performed based on previous work[26]. Colloids with various initial volume fractions were sucked into capillary tubes. The filled capillaries were sealed with UV-curing glue on glass slides and then centrifuged at 450 rcf for 10 min. The experiments were carried out at 20 and 40 °C using a temperature-controlled centrifuge (Avanti® J-25I; Beckman Coulter). The effective volume fraction $\phi_{eff}$ of each sample was calculated in three steps: (1) the wt% of silica particles was obtained by $wt\%(SiO_2) = wt\%(total) \cdot (m_{SiO_2}/m_{PNIPAM} + m_{SiO_2})$, according to the TGA measurements. (2) $wt\%(SiO_2)$ was converted into the volume fraction of silica particles $\phi_{SiO_2}$ using a silica density of 1.8 g/cm³ [56]. (3) $\phi_{eff} = \phi_{SiO_2} \cdot (R_h(T)/R_{h_0}(T))^3$ was calculated based on the DLS measurements. The sediment packing fraction is the slope of the linear fitting of the normalized sediment height versus the suspension's initial volume fraction. The frictional packing fraction $\phi_m$ was evaluated after the centrifugation and the maximal (or final) packing fraction $\phi_f$ was determined after 10 days for suspensions kept at 20 and 40 °C, respectively (Supplementary Fig. 10).

**Shear-rheology measurements.** The shear-rheology measurements were performed on a rheometer (MCR 302; Anton Paar) at 20 and 40 °C in a cone-and-plate geometry, using a 25-mm diameter stainless steel cone with an angle of 2°. The flow curves were recorded for increasing shear rates from 0.1 to 100 s⁻¹. The suspensions were prepared by centrifugation (2701 rcf, 15 min) to produce the sediments at $\phi_m$. Different amounts of Milli-Q water were added to the different sediments to adjust $\phi$ to the desired volume fraction. The suspensions were then sonicated for 30 min before performing the measurements. The viscosity at fixed shear rate (0.1, 1, and 10 s⁻¹) was measured alternately between 20 and 40 °C with 5 min of equilibrium time before every measurement. In all cases, in order to reset the normal force before measuring, the samples were rejuvenated by applying 1% shear strain oscillation for 1 min.

**Adhesion and friction measurements.** AFM (NanoWizard® NanoOptics; JPK Instruments AG) was used to measure the adhesion and friction forces between a PNIPAM-grafted particle and a PNIPAM-grafted substrate. All the measurements were carried out at 20 and 40 °C in a liquid cell (BiOCell^{TM}; JPK Instruments AG). The adhesion force was determined by measuring the pull-off force in a force-versus-distance curve. 200 curves were recorded across a $20 \times 20$ μm² area with the approach and retraction speed set at 500 nm/s. The preparation of colloidal probes and the details of friction measurements are described in previous work[26]. In this work, we used a miBot micromanipulator (Imina Technologies SA) to affix the colloidal particles onto the tipless cantilevers after applying a small amount of epoxy glue at the end of the cantilever. For the friction measurements, the scan area and scan rate were fixed at 2 μm × 200 nm (512 px × 51 px) and 1 Hz, respectively.

At every scanned area, friction loops were recorded at different applied loads $L$ from 10 to 60 nN. The average friction force $F_{\text{friction}}$ for a given load was obtained by averaging between the trace and retrace curves for all the friction loops.

**Reporting summary**. Further information on research design is available in the Nature Research Reporting Summary linked to this article.

## Data availability

Raw data that support the findings of this study are available from the corresponding author upon reasonable request. Source data are provided with this paper.

## Code availability

Analysis code that support the findings of this study are available from the corresponding author upon reasonable request.

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

## Acknowledgements

We thank Jan Vermant and Vincent Niggel for fruitful discussions. We thank Jan Vermant for rheometer access, Markus Niederberger for TGA access, and Andre Studart for SEM access. C.-P.H., N.D.S, and L.I. acknowledge financial support from the ETH Research Grant ETH-49 16-1.

## Author contributions

Author contributions are defined based on the CRediT (Contributor Roles Taxonomy) and listed alphabetically. Conceptualization: C.-P.H., L.I. Formal analysis: C.-P.H., L.I. Funding acquisition: L.I., N.D.S. Investigation: C.-P.H., J.M. Methodology: C.-P.H., L.I., J.M., S.N.R., N.D.S. Project administration: L.I. Supervision: L.I., J.M., N.D.S. Validation: C.-P.H., J.M., S.N.R. Visualization: C.-P.H., L.I. Writing—original draft: C.-P.H., L.I. Writing—review and editing: C.-P.H., L.I., J.M., S.N.R., N.D.S.

## Competing interests

The authors declare no competing interests.
