## [Peer Review File · Nature Communications]

REVIEWER COMMENTS

Reviewer #1 (Remarks to the Author):

The authors present an interesting paper that elucidates the mechanisms that underlie the rheological behavior of dense particulate suspensions. Specifically, the paper focuses on the effects of particle roughness, friction and adhesion on discontinuous shear thickening. The paper is concisely written, experiments were carefully executed, and the conclusions drawn are supported by the data presented. The interesting aspect in this paper is the use of thermally responsive polymer coatings on the particles to control interparticle adhesion, and the masking and unmasking of underlying particle roughness. As such, the chosen approach allows for the deconvolution of effects (roughness, friction, adhesion) on non-Newtonian flow behavior of dense particulate suspensions. Before publication can be recommended, the authors should address the following issues.

1) A control experiment with rough particles in which the PNIPAM brush length exceeds roughness even at elevated temperature. One would expect such a particle to behave more or less like a smooth particle with PNIPAM coating.

2) When first discussing the friction behavior of smooth PNIPAM coated particles (Fig. 2b), it would be good to already go into more detail why such a large difference in the COF arises between coated particles at 20C and those at 40C. It's not only the additional load due to adhesion (which would lead to a simple offset), but also the fact that there are much stronger interactions between the adhering surfaces, requiring higher shear forces for their separation.

3) In Table 1 it would be good to also present the actual particle diameters and their nominal roughnesses.

4) On Page 3, left column, first paragraph: "At 54%, SM PNIPAM..." The word later should be replaced with latter.

5) To be of interest to the broader audience of Nature Comm., the authors need to provide more motivation for their work; particularly also in how the new insights can be leveraged in specific, important systems, where DST is currently a problem.

Reviewer #2 (Remarks to the Author):

This work uses the raspberry particle approach as well as a PNIPAM-chain grafting to create a uniquely tunable range of conditions for surface interactions of dispersed colloidal particles. This is used to explore the surface interaction influence on the rheology of dense suspensions of these particles, with shear thickening the focus and the onset of discontinuous ST a specific sub-focus. The PNIPAM chains are deswelling at an LCST, and when they deswell, they become adhesive. This allows study of sticky vs bare, swollen vs unswollen, smooth vs rough and some combinations of these. The influence on both the tribological measures of adhesive forces and friction, as well as the bulk property of onset and magnitude of thickening (simple ST or DST is clearly demonstrated. The work certainly is both interesting and very innovative, and the conclusions as well as method are important.

I am in favor of publication.

I found it a rather confusing manuscript, with some need to present the role of $\Delta \phi$ a particularly important point. The very dense figures are not fully optimal and could use some simplification.

The most important point I would make is that some consideration of whether the adhesive force is reversible or not. Under shear, it may be ineffective in holding particles together, but one question is whether the dispersion is different after a shearing that carries it into the thickening range. Some treatment of this point seems needed.

Underlying the entire recent upsurge in work on DST is the need for a force, repulsive in general, to keep surfaces separated. What is the force in your case? Maybe I miss the information but did not really catch this.

A minor but important point for ease of reading is to give the particle sizes and fluid properties in the body of the text rather than only in the methods section.

A few other points that struck me in reading:

1) The line "but nowhere is it more prominent than in dense suspensions of solid particles" is oddly close to the first line of the recent review by Morris (2020 Ann Rev Fluid Mech) where it says "nowhere is it more pronounced than in highly concentrated (or dense) suspensions of solid particles".

That work specifically calls out the need for such tribological connection as the current work employs, so the authors may want to go ahead and make reference to it.

2) I was not sure of why the B is in RB in particle name--can this be stated?

3) The line "Strikingly, for the shear-thinning and CST regimes ... confirming that the hydrodynamic volume fraction is the dominating parameter in describing the rheology."

This is very consistent with the good agreement seen in simulations by Stokesian Dynamics of Phung, Brady & Bossis (JFM 1996) or Foss & Brady (JFM 2000) in comparison with experiments of D'Haene, Mewis & Fuller (Rheol Acta 1993) up to about 50% solids; classical Stokesian Dynamics with no allowance for surface contact does not capture the strong thickening above this fraction in the D'Haene et al. paper. I believe this connection would usefully be made.

Similarly, recent work by Wang, Jamali & Brady (J Rheol) and Jamali & Brady (PRL 2019) has used an asperity interlocking argument to describe the shear thickening, above a threshold stress to push past some repulsion. Some discussion of the consistency with their arguments would be useful.

Reviewer #3 (Remarks to the Author):

The manuscript presents a promising application of a recently studied system of smooth and rough silica suspensions coated with a polymer brush coating whose surface properties can be tuned on demand using temperature via a swelling-deswelling transition. The control over the swelling-deswelling transition provides a unique opportunity to separately study the effects of particle roughness, friction, and adhesion on the suspension rheology. Doing so would be a major contribution that would be worthy of publication in Nature Communications. Unfortunately, the experiments that are presented either do not cleanly isolate the effects of particle friction and adhesion, which are always modified in pairs via the deswelling transition (Figs 2e and 3e), or do not isolate the effects of friction from roughness (Figs 2b,f and 3b,f). As such, the paper does not deliver the promise to disentangle the effects of roughness, friction, and adhesion. As it stands, the paper provides experimental evidence that increasing surface roughness and friction, and friction and adhesion, promotes the effects of discontinuous shear thickening, in agreement with existing literature. For the manuscript to be acceptable for Nature Communication, the authors have to show that it is possible to isolate the effects of roughness, friction and adhesion, and conduct a suite of experiments to measure the corresponding rheological (or other macroscopic)

effects.

To summarize, unless the authors can conduct new experiments that truly disentangle roughness, friction, and adhesion, I do not recommend this paper for publication in Nature Communications. Please see below a list of major and minor improvements that should be made for the paper.

Major issues:

1. While the title reads "Disentangling the roles of roughness, friction and adhesion in discontinuous shear thickening by means of thermo-responsive particles", the experiments do not isolate any one of the roughness, friction or adhesion to provide a clean investigation of how that one surface factor affects the suspension rheology. As such, it can only be conclusively shown that both roughness and friction or friction and adhesion enhance shear thickening. As the authors state in the conclusions section "weak adhesion strengthens DST and the combination of roughness and adhesion exacerbates DST." While this is a nice qualitative result, similar ideas have been reported previously. The experimental results do not live up to the title.

2. The authors also claim that in situ temperature tuning allows "DST to be switched on and off on demand". However, it is unclear how long the system takes to transition between the swelling and deswelling state. The repeatability of the transition beyond the 6 measurements presented in figure 4 is also unexplored.

3. In page 3 paragraph 1, the author stated "Strikingly, for the shear-thinning and CST regimes with up to 51%, both suspensions behave identically, even if the contact properties are radically different, confirming that the hydrodynamic volume fraction is the dominating parameter in describing the rheology." This statement is in direct contradiction to the existing view that that CST, like discontinuous shear thickening, is due to the transition of the interaction from non-rolling to rolling friction, as exemplified in Wyart and Cates theory of shear thickening ref. [6]. To substantiate this paradigm changing claim, the authors must provide more evidence than a single volume fraction which exhibits CST in figure 2f, which in itself is not conclusive in showing which parameter is the "dominant" one. Moreover, for that single curve that exhibits CST, it can be argued that the behavior of the grafted and ungrafted particles are actually different with a difference in the onset strain rate, making this claim even more unsubstantiated.

Minor Issues:

1. It is not clear why the only control parameter mentioned is h/d . The height and asperity spacing relative to the particle diameter should be control parameters too and changing h/d as the authors did in the paper necessarily changes either of the two unmentioned control parameters.

2. The reference point for zero distance is not mentioned in figures 2a and 3a.

3. Axes labels are unclear in 2c and 3c. Please add some white space between the two graphs.

4. The axes range in figure 3e and 3f should be identical for easy comparison.

5. A pertinent reference to include is the paper by Wilson Poon's group on "Role of friction in yielding of adhesive non-Brownian suspensions", which explored the effects of adhesion via experiments and simulations.

6. To disentangle the roles of friction, adhesion, and roughness, fully blown phase diagrams that shows how each factor change the landscape of shear thickening suspension will be very useful.

We thank the reviewers for the careful reading of our manuscript and for the useful comments, which we address in detail below. We report the original comments in italics and add our response in normal font following the phrase "AUTHOR REPLY". We also report the changes made to the manuscript text and their location in the manuscript. Revised text appears highlighted in the resubmitted manuscript. All figure numbers listed here refer to this document. The corresponding new figure numbers have been updated in the manuscript and SI.

Reviewer: 1

Comments:

The authors present an interesting paper that elucidates the mechanisms that underlie the rheological behavior of dense particulate suspensions. Specifically, the paper focuses on the effects of particle roughness, friction and adhesion on discontinuous shear thickening. The paper is concisely written, experiments were carefully executed, and the conclusions drawn are supported by the data presented. The interesting aspect in this paper is the use of thermally responsive polymer coatings on the particles to control interparticle adhesion, and the masking and unmasking of underlying particle roughness. As such, the chosen approach allows for the deconvolution of effects (roughness, friction, adhesion) on non-Newtonian flow behavior of dense particulate suspensions. Before publication can be recommended, the authors should address the following issues.

AUTHOR REPLY:

We thank the Reviewer for the positive evaluation of our work.

1) A control experiment with rough particles in which the PNIPAM brush length exceeds roughness even at elevated temperature. One would expect such a particle to behave more or less like a smooth particle with PNIPAM coating.

AUTHOR REPLY:

The control experiment suggested by Reviewer 1 is indeed very important to support our arguments even further. To this end, we have synthesized new rough particles with a roughness

value $h/d = 0.22 \pm 0.03$ and applied SI-ATRP to graft PNIPAM brushes with similar length compared to other three PNIPAM-grafted rough particles (see Table I in revised manuscript). The new particles, RB_0.22_PNIPAM, and the counter surface, RS_0.22_PNIPAM are shown in Figure 1 in this letter.

FIG. 1. Scanning electron microscope images of RB_0.22_PNIPAM (a) and RS_0.22_PNIPAM (b). Scale bars indicate 500 nm.

FIG. 2. Friction-force-vs-applied-load measurements of RB_0.22_PNIPAM, RB_0.36_PNIPAM, RB_0.46_PNIPAM, and RB_0.52_PNIPAM at 20 °C (blue) and 40 °C (red). Error bars represent the standard deviations across the scan area (51 data points). Friction loops of the RB_0.22_PNIPAM and RB_0.52_PNIPAM system at 30 nN and 20 °C (I); at 60 nN and 20 °C (II); at 30 nN and 40 °C (III); 60 nN and 40 °C (IV).

The friction measurements show that the PNIPAM brush in the RB_0.22_PNIPAM system is always able to mask the underlying surface roughness, both at high and low temperature

and at low and high load. This can be clearly seen from the the friction loops at 20 °C and 40 °C, for a 30 nN and a 60 nN load (Figure 2I–IV). In all cases, smooth friction loops are seen for the RB_0.22_PNIPAM system, compared to the RB_0.52_PNIPAM system, which only shows a smooth loop at low temperatures and low loads. Moreover, at 40 °C, the RB_0.22_PNIPAM system shows the lowest friction forces compared to the other RB_PNIPAM systems.

One can further compare the friction loops at 40 °C and 60 nN for all the PNIPAM-grafted systems as shown in Figure 3. It is clear that the RB_0.36_PNIPAM, RB_0.46_PNIPAM, and RB_0.52_PNIPAM particles show large oscillations in the lateral force caused by underlying surface asperities, while the RB_0.22_PNIPAM particles behave very similarly to the SM_PNIPAM particles.

FIG. 3. Friction loops of the five PNIPAM-grafted systems at 60 nN and 40 °C.

Finally, in accordance with the arguments already presented in the original text, the RB_0.22_PNIPAM system has similar μ and ϕ_m values compared to the other RB_PNIPAM systems at 20 °C, but at 40 °C, the RB_0.22_PNIPAM system shows lower μ and higher ϕ_m , as shown in Figure 4a. As a consequence, RB_0.22_PNIPAM shows pure shear thinning instead of DST compared to others at the same volume fraction as shown in Figure 4b.

FIG. 4. **a**, μ as a function of ϕ_m of the four PNIPAM-grafted rough systems (same legends as Figure 2) at 20 °C (blue) and 40 °C (red). μ at 20 °C was calculated from 10 to 30 nN. **b**, Rheological flow curves of the four PNIPAM-grafted rough systems with $\phi = 63\%$ at 20 °C (blue) and $\phi = 46\%$ at 40 °C (red).

2) When first discussing the friction behavior of smooth PNIPAM coated particles (Fig. 2b), it would be good to already go into more detail why such a large difference in the COF arises between coated particles at 20C and those at 40C. It's not only the additional load due to adhesion (which would lead to a simple offset), but also the fact that there are much stronger interactions between the adhering surfaces, requiring higher shear forces for their separation.

AUTHOR REPLY:

We agree with the Reviewer that the increase in the friction at 40 °C for smooth PNIPAM coated particles is not just due to the increased load coming from adhesion, but there is a difference in the dissipation mechanism.

At 20 °C, as seen in the top-left insert of the revised Figure 2a, the polymer chains

are stretched away from the surface in a brush-like conformation presenting strong steric repulsion, and these brushes are lubricious when sliding against each other. At 40 °C, as seen in the top-right insert of the revised Figure 2a, the chains are dehydrated into a collapsed conformation, which significantly decreases the steric repulsion against the opposing surfaces. The conformation change results in non-lubricious sliding as shown in the friction loops in revised Figure 2c (III, IV).

Also, the hydrophobic interactions between the collapsed chains of the opposing surfaces causes more dissipation upon sliding. This increase in the friction when the PNIPAM chains are collapsed was also seen previously [1, 2].

We have modified the discussion of the friction behavior for smooth PNIPAM-coated particles in the revised manuscript as:

”...We then measured the friction by means of lateral force microscopy (see Methods). The friction forces above the LCST are 20 times higher than below the LCST, as shown in Figure 2c and a non-zero friction force at zero load is found, in agreement with the presence of adhesion.. At 20 °C, the polymer chains are stretched away from the surface into a brush-like conformation. The combination of osmotic pressure within the grafted assemblies, and steric resistance to interpenetration between opposing brushes, leads to low friction when sliding against each other. At 40 °C, the chains are dehydrated into a collapsed conformation, which presents a much lower osmotic and steric repulsion between the opposing surfaces, resulting in non-lubricious sliding. The hydrophobic interactions between the collapsed chains of the opposing surfaces also cause more dissipation while sliding.”

3) *In Table 1 it would be good to also present the actual particle diameters and their nominal roughnesses.*

AUTHOR REPLY:

We have now included the hydrodynamic particle radii and their underlying nominal roughnesses in a modified version of Table I.

4) *On Page 3, left column, first paragraph: “At 54%, SM PNIPAM. . . .” The word later should be replaced with latter.*

AUTHOR REPLY:

We thank the Reviewer for pointing out the typo and we have changed it.

5) To be of interest to the broader audience of Nature Comm., the authors need to provide more motivation for their work; particularly also in how the new insights can be leveraged in specific, important systems, where DST is currently a problem.

AUTHOR REPLY:

We thanks the Reviewer for the comment and we have modified the introduction as follows to include a broader description of the implications of DST in the processing of dense suspensions for materials applications:

”...Based on this knowledge, an engineering-tribology approach can be utilized to design the thickening of suspensions. This approach offers new strategies for the design of energy-absorbing materials [3, 4], medical devices [5], and flexible body armors [6–8]. The ability to predict and determine the onset of shear thickening and its severity is also critical for the 3D printing of composite inks [9] containing high solids loadings, such as in the case of ceramic or conductive inks carrying metallic microparticles, e.g., in the ink-jet printing of solder balls for the microelectronics industry [10].”

and

” ...yet been addressed. External control of shear-thickening response has been realized, for instance, by mechanical triggers [11], i.e. vibrations, but an adaptation of the inter-particle interactions via easily tunable variables, such as temperature control in a printer’s nozzle, presents practical advantages.”

Reviewer: 2

Comments:

This work uses the raspberry particle approach as well as a PNIPAM-chain grafting to create a uniquely tunable range of conditions for surface interactions of dispersed colloidal particles. This is used to explore the surface interaction influence on the rheology of dense suspensions of these particles, with shear thickening the focus and the onset of discontinuous ST a specific sub-focus. The PNIPAM chains are deswelling at an LCST, and when they deswell,

they become adhesive. This allows study of sticky vs bare, swollen vs unswollen, smooth vs rough and some combinations of these. The influence on both the tribological measures of adhesive forces and friction, as well as the bulk property of onset and magnitude of thickening (simple ST or DST is clearly demonstrated. The work certainly is both interesting and very innovative, and the conclusions as well as method are important. I am in favor of publication.

AUTHOR REPLY:

We thank the Reviewer for the positive evaluation of our work.

I found it a rather confusing manuscript, with some need to present the role of $\Delta\phi$ a particularly important point. The very dense figures are not fully optimal and could use some simplification.

AUTHOR REPLY:

We thank the Reviewer for the suggestion and have modified the text and figures in the revised manuscript. In particular, we have now introduced $\Delta\phi^*$ as:

”However, qualitative differences beyond a rescaling of the volume fraction emerge. In particular, clear variations in the flow curves are noted if we rescale the volume fraction with respect to ϕ_m for the different systems, introducing $\Delta\phi^* = (\phi_m - \phi)/\phi_m$. This rescaling specifically shows the difference between the volume fraction ϕ at which the rheology is measured and the corresponding value of ϕ_m . It has recently been shown that this rescaled quantity can be correlated with the strength of shear thickening in colloidal suspensions [12].”

The observations that we report in the manuscript in particular align with the last statement and show that by comparing systems at similar values of $\Delta\phi^*$ one can comment on the propensity to show shear thickening in comparable situations.

The most important point I would make is that some consideration of whether the adhesive force is reversible or not. Under shear, it may be ineffective in holding particles together, but one question is whether the dispersion is different after a shearing that carries it into the thickening range. Some treatment of this point seems needed.

AUTHOR REPLY:

The Reviewer brings up an important point. Even if the specific question they ask cannot be directly addressed, in the sense that we have no direct information on the internal structure of the suspension after DST, there are some pieces of evidence that, at least under the experimental conditions accessible in this work, reversibility is present.

1) The silica-PNIPAM particles also remain colloidally stable at 40 °C. For instance, we have performed dynamic light-scattering experiments to determine the hydrodynamic radius of the RB_0.52_PNIPAM particle for cycles of temperatures below and above the LCST of PNIPAM. The results show that the size change is fully reversible over more than 10 measurements, as shown in Figure 5. Corresponding measurements of the zeta potential at 40 °C give a value of approximately -25 mV, originating from residual charges on the silica surface, which are sufficient to prevent aggregation in the absence of shear.

FIG. 5. The hydrodynamic radii (R_h) of the RB_0.52_PNIPAM particles at at 20 °C (blue) and 40 °C (red).

2) We show that the shear response can be fully rejuvenated by changing the temperature in the system (equilibration time between subsequent measurements of 5 minutes) or by repeating the same flow curves in succession. In Figure 6 we show 2 cycles of 4 flow curves for the same dense suspension of RB_0.52_PNIPAM at 20 °C and 40 °C demonstrating that the rheology is fully reproducible even after taking the suspension into the DST regime.

FIG. 6. Two cycles of the flow curves of RB_0.52_PNIPAM systems with $\phi = 63\%$ at 20 °C (blue) and $\phi = 46\%$ at 40 °C (red).

We have now added a comment on these observations in the manuscript.

”Measurements of the hydrodynamic radii of the particles at 20 °C and 40 °C over multiple cycles indicate full reversibility of the particle swelling-deswelling and show that particles remain colloidally stable even at high temperatures.”

”Cycling the temperature above and below the PNIPAM’s LCST enables the complete recovery of the suspension. Similarly, repeating flow curves at 40 °C shows an identical response, indicating that suspensions entering the DST regime can be fully rejuvenated by shear.”

Underlying the entire recent upsurge in work on DST is the need for a force, repulsive in general, to keep surfaces separated. What is the force in your case? Maybe I miss the information but did not really catch this.

AUTHOR REPLY:

In our system, the brush-like conformation of the swollen PNIPAM leads to steric repulsion at 20 °C, which keeps the silica surfaces separated until a critical load is exceeded and brush interpenetration causes an increase in dissipation (e.g. see Figure 2-II). Residual charges on the silica surface offer an electrostatic repulsive force at 40 °C when the PNIPAM brushes

are deswollen, as mentioned above.

We have clarified these issues in the revised text.

A minor but important point for ease of reading is to give the particle sizes and fluid properties in the body of the text rather than only in the methods section.

AUTHOR REPLY:

We thank the Reviewer for the feedback and have included these information in the modified manuscript.

A few other points that struck me in reading:

1) The line "but nowhere is it more prominent than in dense suspensions of solid particles" is oddly close to the first line of the recent review by Morris (2020 Ann Rev Fluid Mech) where it says "nowhere is it more pronounced than in highly concentrated (or dense) suspensions of solid particles". That work specifically calls out the need for such tribological connection as the current work employs, so the authors may want to go ahead and make reference to it.

AUTHOR REPLY:

We thank the Reviewer for the feedback. The similarity in the phrasing of that sentence is purely coincidental! However, we appreciate the remark and have cited the corresponding reference, which indeed very clearly frames the general problem.

2) I was not sure of why the B is in RB in particle name—can this be stated?

AUTHOR REPLY:

We used the name RB as a abbreviation of raspberry-like particles. We have stated this naming in the modified manuscript to avoid confusion for the readers.

3) The line "Strikingly, for the shear-thinning and CST regimes ... confirming that the hydrodynamic volume fraction is the dominating parameter in describing the rheology." This is very consistent with the good agreement seen in simulations by Stokesian Dynamics of Phung, Brady & Bossis (JFM 1996) or Foss & Brady (JFM 2000) in comparison with experiments of D'Haene, Mewis & Fuller (Rheol Acta 1993) up to about 50% solids; classical

Stokesian Dynamics with no allowance for surface contact does not capture the strong thickening above this fraction in the D’Haene et al. paper. I believe this connection would usefully be made. Similarly, recent work by Wang, Jamali & Brady (J Rheol) and Jamali & Brady (PRL 2019) has used an asperity interlocking argument to describe the shear thickening, above a threshold stress to push past some repulsion. Some discussion of the consistency with their arguments would be useful.

AUTHOR REPLY:

We thank the Reviewer for the valuable feedback and have included the discussion in the modified manuscript such that:

”...This observation is consistent with the results from classical Stokesian dynamics simulations [13, 14] that show that CST in dense suspensions can be generated by hydrodynamic forces without allowing surface contact. Moreover, in our case CST is coupled with a negative N_1 (Figure S3c), which confirms that hydrodynamic effects are dominant.”

Reviewer: 3

Comments:

The manuscript presents a promising application of a recently studied system of smooth and rough silica suspensions coated with a polymer brush coating whose surface properties can be tuned on demand using temperature via a swelling-deswelling transition. The control over the swelling-deswelling transition provides a unique opportunity to separately study the effects of particle roughness, friction, and adhesion on the suspension rheology. Doing so would be a major contribution that would be worthy of publication in Nature Communications. Unfortunately, the experiments that are presented either do not cleanly isolate the effects of particle friction and adhesion, which are always modified in pairs via the deswelling transition (Figs 2e and 3e), or do not isolate the effects of friction from roughness (Figs 2b,f and 3b,f). As such, the paper does not deliver the promise to disentangle the effects of roughness, friction, and adhesion. As it stands, the paper provides experimental evidence that increasing surface roughness and friction, and friction and adhesion, promotes the effects of discontinuous

shear thickening, in agreement with existing literature. For the manuscript to be acceptable for Nature Communication, the authors have to show that it is possible to isolate the effects of roughness, friction and adhesion, and conduct a suite of experiments to measure the corresponding rheological (or other macroscopic) effects. To summarize, unless the authors can conduct new experiments that truly disentangle roughness, friction, and adhesion, I do not recommend this paper for publication in Nature Communications. Please see below a list of major and minor improvements that should be made for the paper.

AUTHOR REPLY:

We thank the reviewer for the critical reading of the manuscript and for prompting us to perform new experiments. In particular, we have now synthesized a new batch of rough particles where the underlying surface roughness is such that it can be completely masked also when the PNIPAM brush is collapsed, adding an additional piece of evidence that the effect of roughness can be decoupled from adhesion by a suitably designed brush. This said, it is an inevitable consequence of the materials we used that a collapse of the polymer brush leads to a concomitant increase of both adhesion and friction: this is in the very nature of the dissipation mechanism between adhesive surfaces. In contrast, some of us have shown in previous work (Fernandez et al. PRL 2013) that it is possible to have non-adhesive polymer coatings where the role of the boundary lubrication friction coefficient can be isolated and directly related to the onset of DST. The complementary case, i.e., that one of polymer brushes with identical friction but different adhesion properties, is certainly much more difficult to realize. If at all practical, it would require the ad-hoc comparison among different systems and conditions, which would be most likely irrelevant for any realistic application.

Supported by the results from our additional experiments, we therefore stand by the conclusion that the results presented in this paper offer new possibilities to investigate and explore the effect of surface topography, adhesion and friction on a single system with a highly controlled and well-known polymeric coating.

Major issues:

1. *While the title reads “Disentangling the roles of roughness, friction and adhesion in discontinuous shear thickening by means of thermo-responsive particles”, the experiments do*

not isolate any one of the roughness, friction or adhesion to provide a clean investigation of how that one surface factor affects the suspension rheology. As such, it can only be conclusively shown that both roughness and friction or friction and adhesion enhance shear thickening. As the authors state in the conclusions section “weak adhesion strengthens DST and the combination of roughness and adhesion exacerbates DST.” While this is a nice qualitative result, similar ideas have been reported previously. The experimental results do not live up to the title.

AUTHOR REPLY:

Given the response to the general comment above, we agree, that, strictly speaking, the role of friction and adhesion cannot be fully isolated. To avoid misunderstandings, we propose to modify the title to ”Exploring the roles of roughness, friction and adhesion in discontinuous shear thickening by means of thermo-responsive particles”.

2. The authors also claim that in situ temperature tuning allows “DST to be switched on and off on demand”. However, it is unclear how long the system takes to transition between the swelling and deswelling state. The repeatability of the transition beyond the 6 measurements presented in figure 4 is also unexplored.

AUTHOR REPLY:

The Reviewer raises an important point. The time required to swell and deswell a PNIPAM brush by changing the temperature across its LCST depends on the time required to expel the solvent from the brush [15]. As an example, it has been shown that for PNIPAM gels in the tens of microns range, the deswelling can take up to a few hundred milliseconds [16]. Given the thickness of the brushed coating our particles, we do not expect any influence of the swelling/deswelling times in the response of our suspensions. The rate-limiting step in the switching is instead the time required to equilibrate the temperature of the suspension in the shear cell. We always apply an equilibration time of 5 minutes.

We have now added a comment to that sentence clarifying the time scale for the change of solubility of the PNIPAM brushes.

”Given the rapid response of PNIPAM to temperature changes [16], DST can effectively be switched on and off on demand.”

In response to the second part of the question, we have performed dynamic light scattering experiments to determine the hydrodynamic radius of the RB_0.52_PNIPAM particle at switching temperatures. The equilibration time at a given temperature is 5 minutes, which is same as the rheological measurements. The results show that the repeatability of the transition can go more than 10 measurements as shown in Figure 5.

We did not present the thermo-switchable rheological measurements beyond 6 measurements because of the limitation of solvent evaporation in our system. We have observed that the viscosity of our aqueous dense suspension at a given shear rate would increase after 45 minutes of measurement. This increase of viscosity implies a change of volume fraction in the dense suspension due to the water evaporation, which is difficult to control with a solvent trap for aqueous systems over a prolonged time. For this reason, we presented the thermo-switchable rheological measurements up to 6 measurements to ensure the consistency of our measured system. Additionally, in Figure 6 we show 2 cycles of 4 flow curves of the same dense suspension of RB_0.52_PNIPAM at 20 °C and 40 °C. These flow curves confirm the repeatability of the transition in our system.

3. In page 3 paragraph 1, the author stated “Strikingly, for the shear-thinning and CST regimes with up to 51%, both suspensions behave identically, even if the contact properties are radically different, confirming that the hydrodynamic volume fraction is the dominating parameter in describing the rheology.” This statement is in direct contradiction to the existing view that that CST, like discontinuous shear thickening, is due to the transition of the interaction from non-rolling to rolling friction, as exemplified in Wyart and Cates theory of shear thickening ref. [6]. To substantiate this paradigm changing claim, the authors must provide more evidence than a single volume fraction which exhibits CST in figure 2f, which in itself is not conclusive in showing which parameter is the “dominant” one. Moreover, for that single curve that exhibits CST, it can be argued that the behavior of the grafted and ungrafted particles are actually different with a difference in the onset strain rate, making this claim even more unsubstantiated.

AUTHOR REPLY:

Concerning this comment, we believe that there may be a misunderstanding concerning the mechanism behind CST in our suspensions. One established way to access the origin of

CST is to examine the first normal stress difference, N_1 . When hydrodynamic effects are dominant the transition to CST is coupled to $N_1 < 0$, while friction-dominated dilatant DST leads to $N_1 > 0$. We have shown in Figure S3c that in our case CST is coupled with a negative N_1 , which confirms that hydrodynamic effects are dominant. Moreover, the relevance and consistency of our statement with existing literature was clearly highlighted by point 3 of Reviewer 2, who additionally asked us to further comment on this observation in the manuscript.

The small difference in the onset strain rate (factor of 1.6) between grafted and ungrafted particles can hardly be considered relevant within experimental uncertainties, especially given that they were carried out on different rheometers. Even repetitions of flow curves of the same shear-thickening system measured with the same rheometer would show differences in the onset stress within a factor of 2, e.g., see [17, 18].

Minor Issues:

1. *It is not clear why the only control parameter mentioned is h/d . The height and asperity spacing relative to the particle diameter should be control parameters too and changing h/d as the authors did in the paper necessarily changes either of the two unmentioned control parameters.*

AUTHOR REPLY:

The five PNIPAM-grafted particles we have investigated in this work have the same core particle diameter. Thus h/d can be used to compare roughness between these samples. However, in more general terms, the Reviewer is right that the composite role of particle size, asperity height and distance is important when looking at particles of different core sizes, where not just variations in the asperity density, but also in the total number of asperities may become important. We have added a sentence to clarify this point that: " h/d directly reflects the linear density of asperities with height h and can be used to compare roughness between the PNIPAM-grafted particles investigated in this work since they have the same core particle diameter and all asperities have a spherical cap morphology."

2. *The reference point for zero distance is not mentioned in figures 2a and 3a.*

AUTHOR REPLY:

The zero-distance point is determined as the point at where the approach curve shows non-zero vertical deflection due to interactions with the substrate. We have added this statement to the manuscript.

3. Axes labels are unclear in 2c and 3c. Please add some white space between the two graphs.

AUTHOR REPLY:

Thanks. We have modified the new Figures for a clearer presentation.

4. The axes range in figure 3e and 3f should be identical for easy comparison.

AUTHOR REPLY:

Thanks. We have modified the new Figure for an easier comparison.

5. A pertinent reference to include is the paper by Wilson Poon's group on "Role of friction in yielding of adhesive non-Brownian suspensions", which explored the effects of adhesion via experiments and simulations.

AUTHOR REPLY:

We thank the Reviewer for pointing this out and we have now included the corresponding reference in the revised manuscript.

6. To disentangle the roles of friction, adhesion, and roughness, fully blown phase diagrams that shows how each factor change the landscape of shear thickening suspension will be very useful.

AUTHOR REPLY:

We agree with the reviewer that this would be indeed a very useful thing to have, but it can be hardly called a "minor issue". As we show in the text, the values of adhesive forces obtained in the experiments are rather close to each other, because they are determined by the chemistry of the polymer brush. Expanding this window would require synthesising

a whole new set of particles with different polymer brushes. A similar conclusion can be made for the tailoring of the friction coefficient. As such, we believe that this request lies beyond the scope of the current manuscript and would be best addressed in a suite of new experiments.

-
- [1] Yunlong Yu, Bernard D. Kieviet, Fei Liu, Igor Siretanu, Edit Kutnyanszky, G. Julius Vancso, and Sissi de Beer. Stretching of collapsed polymers causes an enhanced dissipative response of pnipam brushes near their lcst. *Soft Matter*, 11:8508–8516, 2015.
 - [2] S. N. Ramakrishna, M. Cirelli, M. Divandari, and E. M. Benetti. Effects of lateral deformation by thermoresponsive polymer brushes on the measured friction forces. *Langmuir*, 33(17):4164–4171, 2017.
 - [3] O. E. Petel and A. J. Higgins. Shock wave propagation in dense particle suspensions. *Journal of Applied Physics*, 108(11), 2010.
 - [4] F. Pinto and M. Meo. Design and manufacturing of a novel shear thickening fluid composite (stfc) with enhanced out-of-plane properties and damage suppression. *Applied Composite Materials*, 24(3):643–660, 2017.
 - [5] Pickard T. Jonathan, Day. Simon and Haydn. Williams. Surgical and medical garments and materials incorporating shear thickening fluids, 2008: US20090255023A1.
 - [6] Y. S. Lee, E. D. Wetzel, and N. J. Wagner. The ballistic impact characteristics of kevlar (r) woven fabrics impregnated with a colloidal shear thickening fluid. *Journal of Materials Science*, 38(13):2825–2833, 2003.
 - [7] A. Majumdar, B. S. Butola, and A. Srivastava. Optimal designing of soft body armour materials using shear thickening fluid. *Materials and Design*, 46:191–198, 2013.
 - [8] C. D. Cwalina, C. M. McCutcheon, R. D. Dombrowski, and N. J. Wagner. Engineering enhanced cut and puncture resistance into the thermal micrometeoroid garment (tmg) using shear thickening fluid (stf) - armor (tm) absorber layers. *Composites Science and Technology*, 131:61–66, 2016.
 - [9] D. Kokkinis, M. Schaffner, and A. R. Studart. Multimaterial magnetically assisted 3d printing of composite materials. *Nature Communications*, 6, 2015.

- [10] S. Y. Wu, C. Yang, W. Hsu, and L. Lin. 3d-printed microelectronics for integrated circuitry and passive wireless sensors. *Microsystems and Nanoengineering*, 1, 2015.
- [11] N. Y. C. Lin, C. Ness, M. E. Cates, J. Sun, and I. Cohen. Tunable shear thickening in suspensions. *Proceedings of the National Academy of Sciences of the United States of America*, 113(39):10774–10778, 2016.
- [12] Shravan Pradeep, Alan R. Jacob, and Lilian C. Hsiao. Jamming distance dictates colloidal shear thickening, 2020.
- [13] Thanh N. Phung, John F. Brady, and Georges Bossis. Stokesian dynamics simulation of brownian suspensions. *Journal of Fluid Mechanics*, 313:181–207, 1996.
- [14] DAVID R. FOSS and JOHN F. BRADY. Structure, diffusion and rheology of brownian suspensions by stokesian dynamics simulation. *Journal of Fluid Mechanics*, 407:167–200, 2000.
- [15] Jinhwan Yoon, Shengqiang Cai, Zhigang Suo, and Ryan C. Hayward. Poroelastic swelling kinetics of thin hydrogel layers: comparison of theory and experiment. *Soft Matter*, 6:6004–6012, 2010.
- [16] Ahmed Mourran, Hang Zhang, Rostislav Vinokur, and Martin Moeller. Soft microrobots employing nonequilibrium actuation via plasmonic heating. *Advanced Materials*, 29(2):1604825, 2017.
- [17] N. M. James, E. Han, R. A. L. de la Cruz, J. Jureller, and H. M. Jaeger. Interparticle hydrogen bonding can elicit shear jamming in dense suspensions. *Nat Mater*, 17(11):965–970, 2018.
- [18] N. M. James, C. P. Hsu, N. D. Spencer, H. M. Jaeger, and L. Isa. Tuning interparticle hydrogen bonding in shear-jamming suspensions: Kinetic effects and consequences for tribology and rheology. *Journal of Physical Chemistry Letters*, 10(8):1663–1668, 2019.

REVIEWER COMMENTS

Reviewer #1 (Remarks to the Author):

The authors have addressed this reviewers comments sufficiently. The paper is now ready for publication.

Reviewer #2 (Remarks to the Author):

The response to the review of the first version was thorough, and the points of confusion I found are removed. I am happy to recommend publication.

Two comments from this reading:

1) Something is redundant in this text: "is also sufficient to mask the topography of the underlying particle surface for the raspberry-like particles with the lowest roughness, for the batch of raspberry-like particles with the lowest roughness, i.e., "RB 0.22 PNIPAM".

2) p. 3-4: "The hydrophobic interactions between the collapsed chains of the opposing surfaces also cause more dissipation while sliding." I do not follow how these interactions cause more dissipation.

Reviewer #3 (Remarks to the Author):

The authors have made an extensive effort to further support their study by adding results for the RB_0.22 particles and have provided useful clarifications for some of the issues we raised. Importantly, the authors addressed the key concern that the paper fails to deliver what was promised in the title and abstract by retracting the claim of having independent control over roughness, adhesion, and friction. However, given the retraction, it is not clear what is the significant advancement in the understanding of the field. In some sense, the paper now stands at the crossroads between a novel technical application and a system where it may be possible to achieve new fundamental understanding in future experiments. It is not at all clear that this is a sufficient enough advance for Nature Communications.

Major issue:

1. The authors mentioned in their response to the general comment that "it is an inevitable consequence of the materials we used that a collapse of the polymer brush leads to a concomitant increase of both adhesion and friction: this is in the very nature of the dissipation mechanism between adhesive surfaces." It is exactly because the effects are so highly intertwined that the initial promise – that the triad of effects can be independently probed – makes the purported study so appealing. There has been a growing interest in the field in studying these effects, with many recent publications that attempt to study this complicated triad of effects, such as refs [1]-[4] provided below, which have all fallen short of being able to disentangle all 3 effects. Having retracted the initial promise, the paper fails to sufficiently distinguish itself from existing publications. In addition, as the authors mentioned in their response, the effect of polymer masks on surface roughness has already been demonstrated in Fernandez et al. PRL 2013, and the effect of adhesion was explored, but not modified, in Richards J et al (reference 44. which is both an experimental and numerical study of the role of adhesion. The authors also do not put forward any further hypotheses pertaining to shear thickening that can be tested by their system. Thus, it is not clear what new fundamental understanding has been gained.

Minor issues:

1. The premise of the previous version is still held in page 1, last paragraph, line 3 "allows us to test the roles of friction, adhesion, and surface roughness independently ...". As mentioned in the previous response, the study does not offer such independent control.
2. The authors have showed that the RB_0.22 particles have similarity with the SM_PNIPAM particles. It would be instructive to compare the rheology of the two suspensions at 40oC and 20oC, at the same $\Delta\phi$.
3. If the goal of figure 3c is to compare the effect of the volume fraction, why are the comparisons between the curves made at the same $\Delta\phi$ (as opposed to ϕ)?
4. Keeping the axes consistent between the center panel of figure 2c and 4c will make comparison easier.

References:

- [1] Richards, James A., Rory E. O'Neill, and Wilson CK Poon. "Turning a yield-stress calcite suspension into a shear-thickening one by tuning inter-particle friction." arXiv preprint arXiv:2007.05433 (2020).
- [2] James, Nicole M., et al. "Interparticle hydrogen bonding can elicit shear jamming in dense suspensions." *Nature materials* 17.11 (2018): 965-970.
- [3] Singh, Abhinendra, et al. "Shear thickening and jamming of dense suspensions: the roll of friction." *Phys. Rev. Lett.* 124, 248005 (2020)
- [4] Bourrienne, Philippe, et al. "Unifying disparate experimental views on shear-thickening suspensions." arXiv preprint arXiv:2001.02290 (2020).

We thank the reviewers for the careful reading of our manuscript and for the useful comments, which we address in detail below. We report the original comments in italics and add our response in normal font following the phrase "AUTHOR REPLY". We also report the changes made to the manuscript text and their location in the manuscript. Revised text appears highlighted in the resubmitted manuscript.

Reviewer: 1

Comments:

The authors have addressed this reviewers comments sufficiently. The paper is now ready for publication.

AUTHOR REPLY:

We thank the Reviewer for the positive evaluation of our work and the recommendation for publication.

Reviewer: 2

Comments:

The response to the review of the first version was thorough, and the points of confusion I found are removed. I am happy to recommend publication.

AUTHOR REPLY:

We thank the Reviewer for the positive evaluation of our work and the recommendation for publication.

Two comments from this reading:

1) Something is redundant in this text: "is also sufficient to mask the topography of the underlying particle surface for the raspberry-like particles with the lowest roughness, for the batch of raspberry-like particles with the lowest roughness, i.e., "RB_0.22_PNIPAM".

AUTHOR REPLY:

We thank the Reviewer for spotting this mistake. We have modified the text to "is also sufficient to mask the topography of the underlying particle surface for the raspberry-like particles with the lowest roughness, i.e., "RB_0.22_PNIPAM"."

2) p. 3-4: *"The hydrophobic interactions between the collapsed chains of the opposing surfaces also cause more dissipation while sliding." I do not follow how these interactions cause more dissipation.*

AUTHOR REPLY:

The hydrophobic interactions between the collapsed chains make the surfaces adhesive in aqueous solutions. Hysteresis in adhesion causes an increased dissipation, i.e., higher lateral friction, while sliding as described, e.g., in Isrealachvili's book [1]. We have added the reference and modified the sentence in the text to: "The hydrophobic interactions between the collapsed chains of the opposing surfaces also cause more dissipation while sliding, due to adhesion hysteresis."

Reviewer: 3

Comments:

The authors have made an extensive effort to further support their study by adding results for the RB_0.22 particles and have provided useful clarifications for some of the issues we raised. Importantly, the authors addressed the key concern that the paper fails to deliver what was promised in the title and abstract by retracting the claim of having independent control over roughness, adhesion, and friction. However, given the retraction, it is not clear what is the significant advancement in the understanding of the field. In some sense, the paper now stands at the crossroads between a novel technical application and a system where it may be possible to achieve new fundamental understanding in future experiments. It is not at all clear that this is a sufficient enough advance for Nature Communications.

AUTHOR REPLY:

We thank the Reviewer for challenging us to strengthen the conclusions of our study. Our

modification of the title and abstract was not so much a retraction but rather a way to highlight the fact that a true disentanglement of these effects from an experimental viewpoint is practically impossible, as we argue in more detail below.

We stand by our statement that our experimental findings demonstrate the first shear-thickening system where the interparticle roughness, friction, and adhesion can be regulated while the suspensions are being sheared, as a means to provide external control over the global flow properties triggered by modifications of the microscopic contacts. Under the currently developing constraint-based approach to shear thickening, we are confident that our experimental study will stimulate and encourage future simulation studies to include roughness, friction, and adhesion in one system. We also foresee the development of further materials exhibiting smart flow control through contact engineering, inspired by our findings.

Major issues:

1. *The authors mentioned in their response to the general comment that “it is an inevitable consequence of the materials we used that a collapse of the polymer brush leads to a concomitant increase of both adhesion and friction: this is in the very nature of the dissipation mechanism between adhesive surfaces.” It is exactly because the effects are so highly intertwined that the initial promise – that the triad of effects can be independently probed – makes the purported study so appealing. There has been a growing interest in the field in studying these effects, with many recent publications that attempt to study this complicated triad of effects, such as refs [2–5] provided below, which have all fallen short of being able to disentangle all 3 effects. Having retracted the initial promise, the paper fails to sufficiently distinguish itself from existing publications. In addition, as the authors mentioned in their response, the effect of polymer masks on surface roughness has already been demonstrated in Fernandez et al. PRL 2013, and the effect of adhesion was explored, but not modified, in Richards J et al. (reference 44. which is both an experimental and numerical study of the role of adhesion). The authors also do not put forward any further hypotheses pertaining to shear thickening that can be tested by their system. Thus, it is not clear what new fundamental understanding has been gained.*

AUTHOR REPLY:

We would like to provide additional clarification and point out that the intertwining of

friction and adhesion, as mentioned by the Reviewer, is not just manifested in our system, but it is an intrinsic feature of microscopic contacts exhibiting friction and adhesion. Although in a numerical simulation one can independently tune a friction coefficient and an adhesive force (μ and F_0 , respectively, as defined in the text) to regulate the friction force $F_{friction}$, in experiments this is almost impossible.

It has been well documented that solids with high adhesion normally have a higher friction force both in macroscale as well as in nanoscale contacts [6–10]. Particularly focusing on the nanoscale contacts, increase in the adhesion changes the real area of contact and can potentially change the entire contact mechanical behavior [9, 11, 12]. As the friction force is assumed to be directly proportional to the real area of contact, experimentally, it is almost impossible to have a system in which the adhesive force changes but the real area of contact does not. The connection between these two factors implies that a complete, orthogonal disentanglement between friction and adhesion cannot be reached in a real system. We hope that these arguments fully clarify the implications of our results and help shed light on a very active discussion topic.

In fact, while the role of contact friction in shear-thickening systems has been investigated for the last decade, studies on the role of adhesion have only begun in the last two years. Importantly, the interplay of friction, adhesion and roughness has not been experimentally addressed in one single system, as we do in our manuscript.

In particular, in relation to the articles mentioned by the reviewer:

- The work of Fernandez et al. PRL 2013 [13] investigated how shear thickening is connected to contact friction, where the latter was modified by different polymer coatings. The ability of polymers to mask surface roughness in relation to shear thickening was not present in that study. This work demonstrates the first experimental system in which surface roughness can be tuned *in situ* by the PNIPAM brushes.
- The work of Richards J et al. JoR 2020 [14] and refs [2, 3] focus on modifying adhesion and, as a result, changing the friction. In these systems, surfactants or urea molecules are needed to achieve an irreversible modification. Our work shows that adhesion can be switched reversibly using an external stimulus, offering a new design route for novel applications.

- Our work offers an experimental realization to probe the role of adhesion and roughness in restricting the rolling motion of particles, as discussed in the simulation work of ref [4].
- The work of ref [5] proposes that shape irregularity and hydrogen bonding play critical roles in shear thickening, but they provide no measurements of tribological properties to support their arguments. Instead, we have shown that we can measure and quantify the tribological properties of our system and link them directly to their rheological response. Our work remains one of the very few where these links are experimentally explored.

We have added elements of the arguments above, especially focusing on the interplay between friction and adhesion, to the revised version of the manuscript. We thank the Reviewer again and hope that our additional discussion has contributed to clarify and support our study.

Minor Issues:

1. *The premise of the previous version is still held in page 1, last paragraph, line 3 "allows us to test the roles of friction, adhesion, and surface roughness independently ...". As mentioned in the previous response, the study does not offer such independent control.*

AUTHOR REPLY:

We have removed the word "independently" and added an extended discussion as detailed above.

2. *The authors have showed that the RB_0.22 particles have similarity with the SM_PNIPAM particles. It would be instructive to compare the rheology of the two suspensions at 40 °C and 20 °C, at the same $\Delta\phi$.*

AUTHOR REPLY:

We appreciate the Reviewer’s suggestion, but as they mention, we have already shown the similarity between RB_0.22_PNIPAM and SM_PNIPAM. Based on the current data presented in the manuscript, we do not think performing such an experiment would offer further insights into this study.

3. *If the goal of figure 3c is to compare the effect of the volume fraction, why are the comparisons between the curves made at the same $\Delta\phi$ (as opposed to ϕ)?*

AUTHOR REPLY:

The comparisons between the curves in Figures 3c, as well as the corresponding discussion in the text, are already made at the same ϕ .

4. *Keeping the axes consistent between the center panel of figure 2c and 4c will make comparison easier.*

AUTHOR REPLY:

We would rather keep the center panel of Figure 2c as it is, in order to illustrate the difference in friction at 20 °C between SM and SM_PNIPAM at higher loads more clearly.

-
- [1] Israelachvili, J. *Intermolecular and Surface Forces*. Academic Press (Academic Press, 2011).
 - [2] Richards, J. A., O’Neill, R. E. & Poon, W. C. K. Turning a yield-stress calcite suspension into a shear-thickening one by tuning inter-particle friction (2020). arXiv preprint arXiv:2007.05433.
 - [3] James, N. M., Han, E., de la Cruz, R. A. L., Jureller, J. & Jaeger, H. M. Interparticle hydrogen bonding can elicit shear jamming in dense suspensions. *Nat Mater* **17**, 965–970 (2018).
 - [4] Singh, A., Ness, C., Seto, R., de Pablo, J. J. & Jaeger, H. M. Shear thickening and jamming of dense suspensions: The ”roll” of friction. *Phys. Rev. Lett.* **124**, 248005 (2020).
 - [5] Bourriane, P., Niggel, V., Polly, G., Divoux, T. & McKinley, G. H. Unifying disparate experimental views on shear-thickening suspensions (2020). arXiv preprint arXiv:2001.02290.
 - [6] McFarlane, J. S. & Tabor, D. Relation between friction and adhesion. *Proceedings of the Royal Society of London. Series A, Mathematical and Physical Sciences* **202**, 244–253 (1950).

- [7] Bowden, F., Bowden, F. & Tabor, D. *The Friction and Lubrication of Solids*. No. v. 1 in International series of monographs on physics (Clarendon Press, 2001).
- [8] Ramakrishna, S. N., Nalam, P. C., Clasohm, L. Y. & Spencer, N. D. Study of adhesion and friction properties on a nanoparticle gradient surface: Transition from jkr to dmt contact mechanics. *Langmuir* **29**, 175–182 (2013).
- [9] Zhang, Z. *et al.* Nanoscale contact mechanics of biocompatible polyzwitterionic brushes. *Langmuir* **29**, 10684–10692 (2013).
- [10] Gunnewiek, M. K. *et al.* Stem-cell clinging by a thread: Afm measure of polymer-brush lateral deformation. *Advanced Materials Interfaces* **3**, 1500456 (2016).
- [11] Raftari, M., Zhang, Z. J., Carter, S. R., Leggett, G. J. & Geoghegan, M. Nanoscale contact mechanics between two grafted polyelectrolyte surfaces. *Macromolecules* **48**, 6272–6279 (2015).
- [12] Dehghani, E. S. *et al.* Fabrication and interfacial properties of polymer brush gradients by surface-initiated cu(0)-mediated controlled radical polymerization. *Macromolecules* **50**, 2436–2446 (2017).
- [13] Fernandez, N. *et al.* Microscopic mechanism for shear thickening of non-brownian suspensions. *Physical Review Letters* **111**, 108301 (2013).
- [14] Richards, J. A. *et al.* The role of friction in the yielding of adhesive non-brownian suspensions. *Journal of Rheology* **64**, 405–412 (2020).

REVIEWERS' COMMENTS

Reviewer #3 (Remarks to the Author):

I thank the authors for doing their best to clarify their achievements. As the author's themselves state, they have not disentangled these effects, and it is unlikely that it will be possible to do so in an experimental system. I therefore stand by my previous conclusion that the paper while interesting fails to sufficiently distinguish itself sufficiently from existing publications to warrant publication in Nature Communications

REVIEWERS' COMMENTS

Reviewer #3 (Remarks to the Author):

I thank the authors for doing their best to clarify their achievements. As the author's themselves state, they have not disentangled these effects, and it is unlikely that it will be possible to do so in an experimental system. I therefore stand by my previous conclusion that the paper while interesting fails to sufficiently distinguish itself sufficiently from existing publications to warrant publication in Nature Communications.

We thank the reviewer for appreciating our efforts in clarifying our achievements. Even if, in the end, we maintain different opinions on the novelty and impact of the advances offered by our findings, we do thank them for pushing and challenging us into a very constructive review of our work. It is not always possible to reach a full agreement and scientific progress requires constant scrutiny and open discussions.